# Comparison of Different Sequential Assimilation Algorithms for Satellite-derived Leaf Area Index Using the Data Assimilation Research Testbed (Version Lanai)

Xiao-Lu Ling [1,2,3], Cong-Bin Fu [1,2,*], Zong-Liang Yang [3,*], and Wei-Dong Guo [1,2]

5 [1]Institute for Climate and Global Change Research & School of Atmospheric Sciences, Nanjing University, Nanjing 210023, China

[2]Joint International Research Laboratory of Atmospheric and Earth System Sciences of Ministry of Education, Nanjing 210023, China

[3]Department of Geological Sciences, John A. and Katherine G. Jackson School of Geosciences, 10 University of Texas at Austin, Austin, TX 78705, USA

*Correspondence to:* Cong-Bin Fu (fcb@nju.edu.cn); Zong-Liang Yang (liang@jsg.utexas.edu)

**Abstract.** The leaf area index (LAI) is a crucial parameter for understanding the exchanges of mass and energy between terrestrial ecosystems and the atmosphere. In this study, the Data Assimilation Research Testbed (DART) has been successfully coupled to the Community Land Model with explicit carbon and 15 nitrogen components (CLM4CN) by assimilating Global Land Surface Satellite (GLASS) LAI data. Within this framework, four sequential assimilation algorithms, including the Kernel Filter (KF), the Ensemble Kalman Filter (EnKF), the Ensemble Adjust Kalman Filter (EAKF), and the Particle Filter (PF), are thoroughly analysed and compared. The results show that assimilating GLASS LAI into the CLM4CN is an effective method for improving model performance. In detail, the assimilation accuracies 20 of the EnKF and EAKF algorithms are better than that of the KF and PF algorithm. From the perspective of average and RMSD, the PF algorithm performs worse than the EAKF and EnKF algorithms because of the gradually reduced acceptance of observations with assimilation steps. In other words, the contribution of the observations to the posterior probability during the assimilation process is reduced. The EAKF algorithm is the best method because the matrix is adjusted at each time step during the 25 assimilation procedure. If all the observations are accepted, the analysed LAI seem to be better than that when some observations are rejected, especially in low-latitude regions.

## 1 Introduction

Land surface processes play an important role in the earth system because all the physical, biochemical, and ecological processes occurring in the soil, vegetation, and hydrosphere influence the mass and energy 30 exchanges during land-atmosphere interactions (Bonan, 1995; Pitman, 2003; Pitman et al., 2009, 2012). The leaf area index (LAI) is a key biophysical parameter of vegetation in land surface models (LSMs) and influences their simulation performance. Therefore, high-quality, spatially and temporally continuous LAI inputs are extremely important (Bonan et al., 1992; Li et al., 2015).

Real-time monitoring of LAI on a large scale is a worldwide problem. The lack of spatial 35 representativeness caused by the sparse distribution of conventional observations makes it difficult to

achieve a global observational LAI dataset. Remote sensing can provide global data with high spatial and temporal resolutions, but the inversion accuracy is associated with different plant functional types (PFTs) and vegetation fractions. Furthermore, although advanced land surface models (LSMs, e.g., the Community Land Model version 4, CLM4) can predict LAI variation, the model performance is greatly

affected by the model structure or the initial/forcing/boundary conditions of the input (Dai et al., 2003; Luo et al., 2003; Levis et al., 2004). Data Assimilation (DA), through optimally combining both dynamical and physical mechanisms with real-time observations, can effectively reduce the estimation uncertainties caused by spatially and temporally sparse observations and poor observed data accuracy (Kalnay, 2003).

As a link between observations and dynamic model states, mathematical algorithms play an important role in calculating the increments and adjusting the state vector during assimilation (Kalnay et al., 2007). The two basic data assimilation algorithms are the variational DA based on optimal control theory and sequential algorithms based on the Kalman Filter (KF) (Dimet and Talagrand, 1986; Gordon et al., 1993; Bannister et al., 2017; Vetra-Carvalho et al., 2018). Because the KF algorithm is based on the linear

model error assumption, many new sequential algorithms have been proposed. For example, the Extended Kalman Filter (EKF) was developed to meet the need for a nonlinear observation operator, but the tangent operator needs to be developed (Kalnay, 2003). Based on the Monte Carlo method and focused on the nonlinear operator, the Ensemble Kalman Filter (EnKF) was developed (Evensen, 1994) and was first used in the study of atmospheric science (Houtekamer and Mitchell, 1998). Since then, the

EnKF has been widely applied for the assimilation of ocean, land surface and atmospheric data (Houtekamer et al., 2005; Evensen, 2007). In recent years, the Monte-Carlo methods have been proposed to allow the assimilation of information from sources that have non-Gaussian errors.

Many previous studies focusing on the comparison of variational and sequential algorithms have been conducted to determine the optimal assimilation method (Han and Li, 2008). Wu et al. (2011)

systematically compared EnKF and 3DVAR/4DVAR algorithms and found that the EnKF algorithm was better than the 3DVAR method and the same as the 4DVAR method. For this reason, the application of the EnKF algorithm has been expanded quickly, and many other forms of the EnKF method have been developed, such as the Dual EnKF (Li et al., 2014), Ensemble Square Root Filter (EnSRF) (Whitaker and Hamill, 2002), and Ensemble Adjust Kalman Filter (EAKF, Anderson, 2001). At the same time,

combinations of variational algorithms and sequential algorithms have also been developed. For example, the maximum likelihood ensemble filter (MLEF, Zupanski, 2005), the combination of 3DVAR and PF algorithms (Leng and Song, 2013), the hybrid variational-ensemble data assimilation methods, i.e., the 4DEnKF (Hunt et al., 2004; Fertig et al., 2007; Zhang et al., 2009) and the DrEnKF (Wan et al., 2009) have been developed at NCEP and applied to improve model predictions (Whitaker et al., 2008).

A complete Land Data Assimilation System (LDAS) is mainly composed of forcing datasets, initial and boundary datasets, parameterization sets, dynamical models as physical constraints, assimilation algorithms, observational data and target output. In recent decades, studies of land data assimilation have become very active, although this topic was proposed later than the assimilation of atmospheric observations (Lahoz and De Lannoy, 2014). Land data assimilation can implement both in-situ

observations and remotely sensed data like satellite observation of soil moisture, snow water equivalent

(SWE), land surface temperature and so on to constrain the physical parametrization and initialization of land surface state. (Liu et al., 2008; Reichle et al., 2014; Zhang et al., 2014; Zhao et al., 2016; 2018). The widely acknowledged LDASs include the North LDAS (NLDAS, Mitchell et al., 2004; NLDAS-2, Luo et al., 2003; Xia et al., 2012), the Global LDAS (GLDAS, Rodell et al., 2004), the European LDAS (ELDAS, Jacobs et al., 2008), the West China LDAS (WCLDAS, Huang and Li, 2004) and the Canadian LDAS (CaLDAS, Carrera et al., 2015).

Recent studies focusing on assimilation in terrestrial systems have tended to add multiple phenological observations to constrain and predict biome variables and further improve model performance (Knyazikhin et al., 1998; Xiao et al., 2009; Viskari et al., 2015). Joint assimilation of surface incident solar radiation, soil moisture and vegetation dynamics (LAI) into land surface models or crop models is of great importance since it can improve the model results for national food policy and security assessments (Sabater et al., 2008; Ines et al., 2013; Sawada et al., 2015; Jin et al., 2018; Mokhtari et al., 2018). Furthermore, the abilities to simulate river discharge, land evapotranspiration, and gross primary production have been improved in Europe (Barbu et al., 2011; Albergel et al., 2017). To date, such studies have been conducted using a single sequential algorithm at a single site or on regional scales (Montzka et al., 2012; Sawada et al., 2018).

The Data Assimilation Research Testbed (DART) is an open source community facility and includes several different types of KF algorithms (Anderson et al., 2009). It has been coupled to many high-order models and observations for ocean, atmosphere, land surface, and chemical constituents. For example, DART has been coupled with CLM4 or CLM4.5 to improve snow and soil moisture estimations as well as land carbon processes (Zhang et al., 2014; Kwon et al., 2016; Zhao et al., 2016; Fox et al., 2018; Zhao et al., 2018). Utilizing coupled DART/CLM4, the Global Land Surface Satellite LAI (GLASS LAI) data are assimilated into the Community Land Model with carbon and nitrogen components (CLM4CN) in the present study to explore the optimal assimilation algorithm for model performance. The experimental design and different assimilation algorithms are described in Sect. 2. Section 3 describes the optimal algorithm for LAI assimilation, and the proportion of observations is discussed in Sect. 4. Conclusions and discussions are given in Sect. 5.

## 2 Data and Methodology

A complete LDAS is mainly composed of forcing/initial/boundary datasets, parameterization sets, dynamical LSMs, assimilation algorithms, observational data and target output. LSMs play an important role in the LDAS because they can add physical constraints to the control variables during assimilation. In addition, the simulation ability of LSMs can directly affect the output because they provide the associated uncertainty for assimilation.

### 2.1 CLM4CN

Developed by the National Center for Atmospheric Research (NCAR), the Community Land Model (CLM) can simulate energy, momentum and water exchanges between the land surface and the overlying atmosphere at each computational grid. The CLM is designed mainly for coupling with the atmospheric

numerical model and providing the surface albedo (direct and scattered light within the visible and infrared bands), upward longwave radiation, sensible heat flux, latent heat flux, water vapor flux, and east-to-west and south-to-north surface stress needed by the atmospheric model. These parameters are controlled by many ecological and hydrological processes. The model can also simulate leaf phenology

and physiological processes, as well as water circulation through plant pores. Ecological differences between vegetation types and thermal and hydrological differences between different soil types are also considered. Each grid cell can be covered by several different land use types. Each cell contains several land units, each land unit contains a different number of soil and snow cylindrical blocks, and each cylindrical block may contain several types of vegetation functions. The CLM employs 10 soil layers to

resolve soil moisture and temperature dynamics and uses PFTs to represent subgrid vegetation heterogeneity (Oleson et al., 2010).

There are two ways to update LAI in CLM4. The LAI is treated as a diagnostic variable that is linearly interpolated from a 30-year averaged satellite dataset, and there is no annual LAI variation for CLM4 with Satellite Phenology (CLM4SP) (Lawrence and Chase, 2007). For CLM4CN, the prognostic LAI is

15 calculated by the leaf carbon pool and an assumed vertical gradient of specific leaf area (SLA) (Thornton and Zimmermann, 2007). Carbon and nitrogen are obtained by plant storage pools in one growing season and then retained and distributed in the subsequent year. All carbon and nitrogen state variables in vegetation, litter, and soil organic matter (SOM) are prognostic based on the prescribed vegetation phenology. The CLM4CN offline mode with prescribed meteorological forcing is used in this study.

**2.2 DART (the Lanai version)**

DART is developed and maintained by the Data Assimilation Research Section (DAReS) at NCAR. The purpose of DART is to provide a flexible tool for data assimilation (DA), and it has been coupled with many 'high-order' models. As a software environment, DART makes it easy to explore a variety of data assimilation methods and observations with different numerical models. The DART system includes

several different types of sequential algorithms, which are selected at runtime by a namelist setting. The Lanai version of DART, which supports many existing models including the CESM climate component, the MPAS (Model for Prediction Across Scales) models and the NOAH land model etc., is used in this study. Released in December 2013, the Lanai version of DART can process many new observation types and sources, and include new diagnostic routines as well as new utilities. Detailed settings for DART can

be found at https://www.image.ucar.edu/DAReS/DART/.

Currently, the coupled DART/CLM4 model has produced many reanalysis data for snow and soil moisture. It has been found that snow DA can improve temperature predictions, especially over the Tibetan Plateau, implying great implications for future land DA and seasonal climate prediction studies (Lin et al., 2016). Furthermore, the coupled DART/CLM framework would be employed to assimilate

other variables, such as LAI, from various satellite sources and ground observations (i.e., truly multi-mission, multi-platform, multi-sensor, multi-source, and multi-scale). Ultimately, this would allow earth system models to be constrained by all types of observations to improve model performance for seasonal and decadal prediction skills.

## 2.3 Sequential Assimilation Algorithms

According to Anderson et al. (2001), Equation (1) is used to express how new sets of observations modify the prior joint state conditional probability distribution obtained from predictions based on previous observation sets.

$$\mathbf{p}(\mathbf{z}_{t,k}|\mathbf{Y}_{t,k}) = \mathbf{p}(\mathbf{y}_{t,k}^{o}| \mathbf{z}_{t,k})\, \mathbf{p}(\mathbf{z}_{t,k} | \mathbf{Y}_{t,k-1})/\, \mathbf{p}(\mathbf{y}_{t,k}^{o}| \mathbf{Y}_{t,k-1}) \qquad\qquad (1)$$

in which $Y_{t,k}$ is defined as the superset of all observation subsets, $\mathbf{y}_{t,k}^{o}$ is the kth subset of observations at time t, $z_{t,k}$ is the joint state-observation vector for a given t and k. In ensemble applications, generally there is no need to compute the denominator of (1). Four algorithms for approximating the product in the numerator of (1) are presented below, and detailed information can be found in Anderson et al. (2001).

### 2.3.1 Kernel Filter (KF)

The Kernel Filter mechanism, first proposed by Lindgren et al. (1993) and further developed by Anderson and Anderson (1999), has been incorporated into the DART and can be extended to the joint state space. Detailed calculation process can be found in Anderson et al. (2001). The KF is potentially general, because the values and expected values of the mean and covariance and higher-order moments of the resulting ensemble are functions of high-order moments of the prior distribution. However, when applied to large models, computational efficiency will be an issue for the application of the algorithm.

### 2.3.2 Ensemble Kalman Filter (EnKF)

The KF algorithm has not been widely used because of computing limitations and the linear model error assumption. The EnKF was proposed based on a Monte Carlo approximation, for which the background error covariance is approximated using an ensemble of forecasts (Evensen, 1994). The EnKF algorithm can be utilized for nonlinear systems and can also reduce the computing requirement of DA (Burgers et al., 1998; Evensen, 2003; 2007).

The EnKF procedure is divided into two stages: prediction and analysis. (1) In the prediction stage, the ensemble forecast field is generated from the ensemble initial condition, and the error covariance matrix of the ensemble forecast is calculated. (2) In the analysis stage, the simulation of each member of the ensemble is updated using the covariance matrix of observation vector error and state vector error. The traditional EnKF, an ensemble of Kalman Filters with each member using a different sample estimate of the prior mean and observations, is used in this study (Houtekamer and Mitchell1998).

### 2.3.3 Ensemble Adjust Kalman Filter (EAKF)

Although the forms of expression are different, the proposed EnSRF (Whitaker et al., 2002) and EAKF (Anderson, 2001) are the same algorithm.

The difference between the EAKF and the traditional EnKF lies in the adjustment of the gain matrix to avoid filtering the divergence problem by increasing the premise of the analysis error covariance (Anderson, 2003, 2007; Wang et al., 2007). In the EAKF algorithm, ensemble observation members are calculated by the observation operator, and the increment of each observation member is calculated as $\Delta Y_i$.

The increment $\Delta X_{ij}$ for each ensemble sample of each state variable in terms of $\Delta Y_i$ can then be calculated as follows:

$$\Delta X_{ij} = \frac{\sigma_{j_o}^p}{\sigma_o^p + \sigma_{j_o}^p} \Delta Y_i. \tag{2}$$

where i indicates the ensemble member, j is the state vector member, $\sigma_{j_o}^p$ is the prior covariance of state vector and observation, and $\sigma_o^p$ is the prior variance of observation.

### 2.3.4 Particle Filter (PF)

The Particle Filter (PF) is also a sequential Monte Carlo method, which is based on the Bayesian sequential importance sampling method (SIS). The PF algorithm finds a set of random samples in the state space to approximate the probability density function and then replaces the integral operation with the sample mean to obtain the process of minimum variance distribution of the state (Moradkhani et al., 2005). The procedure of the PF algorithm can also be divided into two frameworks: forecast and analysis. If there are enough observations, the posterior density at k can be approximated as

$$p(X_k^a|Y_{1:k}) \approx \sum_{i=1}^N w_{i,k} \, \delta(X_k^a - X_{i,k}^a). \tag{3}$$

$\delta(*)$ is the Dirac Function and $\sum_{i=1}^N w_{i,k} = 1$.

in which $p(X_k^a|Y_{1:k})$ is the posterior probability distribution, $X_{i,k}^a$ is the particle element, $w_{i,k}$ is the weight of each particle, N is the number of particles. Unlike the EnKF algorithm, the PF method takes into account the weights of different particles and can be better applied to nonlinear systems. However, in association with the DA, there are a limited number of particles with large weights, and too many computing resources are distributed to particles with weights of approximately 0. This situation is called particle degradation (Doucet et al., 2000). Effective methods to solve this issue include resampling or selecting more reasonable importance functions.

### 2.4 Datasets

### 2.4.1 Ensemble Meteorological Forcing and initial conditions

The ensemble initial conditions and background error (Hu et al., 2014) are produced from ensemble analysis products generated by running DART and the Community Atmosphere Model (CAM4) (Raeder et al., 2012). DART/CAM4 produced 80 atmospheric forcing datasets with 6-hour time intervals for the period of 1998-2010. These ensemble meteorological data have been widely employed in DA for ocean, snow, soil moisture, and many other related studies (Danabasoglu et al., 2012). By considering computational cost and filter performance, 40 members among the ensemble forcing datasets are chosen to drive the CLM4CN.

To achieve a steady state solution for all state variables, the CLM4CN was run for 4000 years by Qian's forcing (Qian et al., 2006) at the resolution of 1.9° latitude by 2.5° longitude (Shi et al., 2013). The CLM4CN was then forced by the ensemble mean of selected 40 members of DART/CAM datasets for 1000 years. In the last step, the ensemble simulation during the time period from 1998 to 2001 was treated

as spin-up process, and 40 ensemble initial conditions were obtained. Aiming at global scale and considering the computational cost, only one-year assimilation and ensemble simulation were conducted. Our goal is to first find out the best experiment, and then conduct long-term simulation or assimilation in the future.

**2.4.2 LAI datasets**

The Global Land Surface Satellite (GLASS) LAI dataset is used in this study as observations for assimilation (Zhao et al., 2013). Since the ensemble simulation or assimilation is run at the resolution of 0.9° latitude by 1.25° longitude, the original spatial resolution of 0.05° of the GLASS LAI is upscaled to the same resolution.

An independent LAI dataset from the Copernicus Global Land Service (CGLS) with version 2 (GEOV2 LAI) was utilized to validate the assimilation result. The GEOV2 LAI is derived from the vegetation instruments on Satellite Pour I'Observation de la Terre (SPOT-VGT) and board of PROBA satellite (PROBA-V satellites) (Verger et al., 2014). The resolution of GEOV2 LAI is 1-km, which is also upscaled to the grid level to evaluate the analysis of LAI and assimilation effect.

**2.5 Experimental Design**

To determine the optimal assimilation algorithm, five experiments corresponding to the KF, EnKF, EAKF and PF methods are designed and showed in Table 1, in which the "Algorithms" experiments would reject some observations under certain conditions using the KF, EnKF, EAKF, and PF algorithms. The expected value of the difference between the prior mean and observation is $\sqrt{\sigma_{prior}^2 + \sigma_{obs}^2}$, in which

$\sigma_{prior}$ and $\sigma_{obs}$ are standard deviations of the prior PDF and observation PDF respectively. DART will reject the observation if the bias of the prior mean and observations is larger than three times of the expected value. The "Algorithms without observation rejection" experiments would accept all the observed LAI.   During assimilation, CLM stops and writes restart and history files at a frequency of 8 days. If there is available observational GLASS LAI data, they are assimilated into the CLM4CN. DART

extract state vector, the increments are calculated by filtering at each time step, and the LAI, leaf carbon (Leaf C) and leaf nitrogen (Leaf N) are updated. The adjusted DART state vector is resent to the CLM restart files as a new initial condition for the next time step. All the simulation and assimilation are conducted at the spatial resolution of 0.9° latitude by 1.25° longitude. The ensemble assimilation is conducted pointwise, indicating that spatial covariances are not considered.

**3 The Optimal Algorithm for DART/CLM4CN**

The spatial distributions of global LAI in 2002 for (a) GEOV2 LAI in July, (b) ensemble mean of simulations in July, (c) GEOV2 LAI in November, and (d) ensemble mean of simulations in November are shown in Fig. 1. The observations in Fig. 1 are from the upscaled GEOV2 LAI dataset with a spatial resolution of 0.9 latitude by 1.25 longitude. There are two latitudinal belts of high LAI values located in

the tropics and at 50-65°N in July. These two regions are mainly dominated by evergreen broadleaf

forests and boreal forests, respectively. There are 3 high-LAI regions located in the tropics: the Amazon, central Africa, and some islands in Southeast Asia. Because of the presence of deserts, plateaus and bare ground, the LAI is low in northern Africa, western North America, western Australia, southern Africa, and southern South America, where shrubs and/or grass are dominant. Globally, the CLM4CN can simulate the LAI distribution characteristics, except that it systematically overestimates LAI, especially at low latitudes and boreal forest regions, with the largest bias of 5 $m^2/m^2$. The global LAI is lower in November than in July. The LAI values in the high latitudes of the northern hemisphere are higher in July than in November because November is not the growing season for most of the vegetation in the northern hemisphere.

The differences between the methods of (a) EAKF, (b) EnKF, (c) KF as well as (d) PF and GEOV2 LAI are displayed in Fig. 2. Globally, the difference between assimilation with the four methods and GEOV2 LAI are larger in lower-latitude regions, indicating that assimilation also overestimate the LAI value in these regions. The biases of assimilation and observation reduce to 2 $m^2/m^2$ in the low latitude regions compared with the biases of simulation and observation in Fig. 1, where are dominated by BET tropical and mixed forest types. The LAI values from the assimilation experiment are always 1 $m^2/m^2$ higher in the middle- and high-latitude regions, especially in western North America, northwestern China, and western Australia, where open shrublands and grasslands are dominant. Assimilation always underestimate the LAI values in the eastern North America, the northeastern China, and the 50-65°N latitude regions of Eurasia, where are dominated by NET boreal forests and mixed forest types. The assimilation with the EAKF and EnKF algorithms display a lower bias than the KF and PF algorithms compared to GEOV2 LAI, especially in the northern and eastern Amazon, central Africa, southern Eurasia, and Southeast Asia. Notably, the correction of overestimated LAI is significantly better than that of underestimated LAI, which is mainly attributed to the high dispersion of LAI in those regions. In other words, high dispersion is beneficial to assimilation.

The results also indicate that the EAKF and EnKF assimilation algorithms are better than the KF and PF algorithms in November (figures not shown). In detail, the EAKF algorithm is better than the EnKF method in November, especially in the Amazon, central Africa, and southern Eurasia. The biases of assimilated LAI relative to the observed LAI are higher in November in the 20-65°N region, which may be because vegetation during this period in the Northern Hemisphere is not lush. In western Australia and central Eurasia, the improvement of the underestimation in November is not as significant as that in July, which indicates that the system has a limited capability to simulate the vegetation process, especially for open shrubland and grassland. From the perspective of the average and RMSE, the PF algorithm performs worse than the EAKF and EnKF algorithms because of the gradually reduced acceptance of observations with assimilation steps (will discuss below). Note that the average and RMSE only make sense for the Ensemble Kalman Filters. For the PF algorithm, the particle with the largest weight (a posteriori maximum for the pdf) should be discussed separately.

The RMSEs of ensemble members are showed in Figure 3 to provide hints where the assimilation is the most efficient. The RMSEs of ensemble members for the EAKF and EnKF algorithm are larger than those for the KF and PF algorithms, indicating that the EAKF and EnKF are more effective. In July 2002, the RMSE of the ensemble estimates is the largest in lower latitude regions, with particularly high values

in central South America, central Africa, and Southeast Asia. The regions with comparatively large ensemble spreads are located in western North America and western Europe. The large ensemble spreads areas are also transitional regions with different vegetation types, indicating low capability of the models to simulate complex vegetation types.

The globally mean LAI and the LAI in five latitudinal bands were chosen for analysis in this study. The five bands are boreal (45-65°N), northern temperate (23-45°N), northern equatorial (0-23°N), southern equatorial (0-23°S), and southern temperate (23-90°S). Figure 4 presents the root mean square deviation (RMSDs) of the ensemble means of simulation/assimilation versus GEOV2 LAI for (a) global, (b) boreal, (c) northern temperate, (d) northern equatorial, (e) southern equatorial, and (f) southern temperate.

Generally, although they all feature similar variation pattern characteristics, the RMSDs of all the assimilation datasets relative to the GEOV2 LAI are less than those of the simulation, indicating that all four assimilation algorithms can improve the LAI estimation. For boreal regions, there are two maxima for the RMSD in May and September respectively, which is also the period with abrupt variation for LAI value. During the growing season, the RMSDs of LAI reach relatively low values, especially for the

regions in the middle and high latitudes of the Northern Hemisphere and high latitudes of the Southern Hemisphere. In the low-latitude region covered by evergreen or deciduous broadleaf forests, the RMSD does not present an obvious annual change. The EnKF algorithm performed best in the boreal region with the smallest RMSD, while not so good in the northern temperate and northern equatorial regions. The EAKF algorithm are presented the lowest RMSD in the southern equatorial and southern temperate

regions, as well as global regions. The assimilation is far less efficient in the boreal region than in other areas, which is partly attributed to the consistently low initial RMSD during non-growing seasons and limited capability of the models for simulating processes associated with boreal forest type.

Figure 5 shows the globally or regionally averaged RMSDs of simulation/assimilation and GEOV2 LAI. The RMSDs of assimilation are lower than those of simulation, implying that assimilating remotely

sensed LAI data into the CLM4CN is an effective method for improving the model performance. The difference between simulation and all four algorithms in the northern and southern equatorial regions is larger than in other regions, indicating that the assimilation is more efficient there. The global averaged RMSD for LAI from the EAKF experiment is lower than the other three algorithms, except for the boreal regions, indicating the better performance in assimilation.

The background/analysis departures are calculated as (1) innovations, which are the differences between the assimilated LAI and model background, and (2) residuals, which are the differences between the assimilated LAI and analysis (Barbu et al., 2011). It was concluded that the LDAS system is working well based on the condition that the residuals are reduced compared to the innovations (Albergel et al.,

2017). Figure 6 shows the histograms of innovation and residuals of LAI globally and for all subregions during July 2002. Generally, the distribution characteristics of both innovations and residuals are similar for the algorithms of KF and PF, which means that these two algorithms are not very efficient for LAI assimilation. The distribution of residuals is more centered on 0 than that of the innovations for the EAKF and EnKF algorithms, especially for the EAKF algorithm. The innovations dominantly exhibit a large

negative bias, indicating that the model always highly overestimates LAI. The residuals can improve this

overestimation situation, especially for the EAKF algorithm. The analysis departures for the EAKF algorithm are more centered on 0 than the EnKF algorithm, especially in global, northern temperate, and southern temperate regions.

**4 Effective Observational Proportion**

The assimilation results depend not only on the algorithm but also on the observations. This not only requires a sufficiently strong degree of discretization for ensemble simulations but also requires the observational variables to be sufficiently trustworthy. In this section, the proportion of LAI observations that can be accepted for the four algorithms is discussed. During assimilation, the DART can calculate the number of non-assimilated observations when the difference of prior mean and observations is larger

than three times of the expected value. The proportion of accepted LAI observations is defined as the number of accepted observations divided by the number of total observations.

To explain the relationship between assimilation algorithms and observation rejection, Fig. 7 displays the proportion of accepted LAI observations for the four algorithms in the zonal regions. In general, the EnKF and EAKF methods accepted many more observational LAI observations than the PF and KF

methods. In the low-latitude regions, the proportion of accepted LAI observations is approximately 75%, which is lower than in the high-latitude regions. This may be because the broadleaf forest in tropical regions can grow unrestrictedly in the model, producing LAI values that are much higher than the observations. At the very beginning of assimilation, DART rejects the largest proportion of LAI observations in the southern equatorial, northern equatorial, and northern temperate zones due to large

biases between the simulation and the observations. Over time, the rejection proportion gradually decreases for the northern equatorial, southern equatorial and southern temperate. As ensemble-analyzed LAI values tend to relatively fixed, the rejection proportion increases over regions with small LAI amplitudes, such as the northern temperate and boreal region. From May to September in the boreal region and from April to September in the northern temperate region, the proportion of accepted LAI is

much smaller than in the other regions. These two periods with abrupt variation for LAI value are also when the model simulation presents an obvious discrete characteristic. This experiment illustrates the utility of the spin-up process for ensemble initial conditions. Furthermore, the KF and PF algorithms gradually reduce the acceptance of observations as assimilation progresses, which may partially explain their worse performance than the EnKF and EAKF algorithms (see Fig. 5).

The difference between globally assimilated and GEOV2 LAI with the methods of EAKF (with rejection) in (a) July and (b) November are shown in Fig. 8 to illustrate the role of observation proportion. It can be concluded that when accepting all the observations, the assimilation results seem to be better than when some observations are rejected during assimilation. Large negative biases occur in the Amazon, central Africa, southern Eurasia, and the boreal region, where the LAI is overestimated in the model.

Large positive biases occur in the southeastern China, the west North America, west Australia, and the central South America in July, partly due to the influence of topography. In November the positive biases are observed around the whole middle and high latitude regions of the northern hemisphere, indicating the overestimation for LAI value in non-growing seasons.

During assimilation, the assimilated observations (GLASS LAI) are always treated as "true" values. The question thus becomes how do the true values influence the assimilation results? Figure 9 shows the RMSDs of simulation experiments without/with rejection (EAKF_noreject / EAKF_reject) and GEOV2 LAI over the (a) global, (b) boreal, (c) northern temperate, (d) northern equatorial, (e) southern equatorial, and (f) southern temperate regions. In the EAKF_reject experimental design, if the observed LAI is three times larger than the bias between the simulation and the observations, the observation would be rejected by DART, while in the EAKF_noreject experiment, all observed LAIs are assimilated. Generally, RMSDs for both simulation and assimilation present obvious annual variations. The RMSD of assimilation is far less than that of the simulation, although their characteristic variation patterns are similar. This demonstrates the effectiveness of assimilation for improving model simulation. The RMSD relative to the observations was highest for the simulation, followed by the EAKF_reject experiment, and was lowest for the EAKF_noreject experiment. During assimilation, when accepting all the observations, the RMSD is smaller than that when rejecting some observations. Compared with EAKF_reject experiment and other algorithms in Fig.5, the globally and regionally averaged RMSDs from the EAKF_noreject experiment is much smaller, indicating the most efficient performance.

## 5 Conclusions and Discussion

The Community Land Model version 4 with prognostic carbon and nitrogen components (CLM4CN) is coupled with the Data Assimilation Research Testbed (DART) to determine the optimal assimilation algorithm for leaf area index (LAI). The Kernel Filter (KF), Ensemble Kalman Filter (EnKF), Ensemble Adjust Kalman Filter (EAKF), and Particle Filter (PF) are discussed in this paper.

The results show that assimilating remotely sensed LAI into the CLM4CN is an effective method for improving model performance. Globally speaking, the EAKF and EnKF assimilation algorithms are better than the KF and PF assimilation algorithms. The LAI obtained by the EAKF algorithm is more continuous than that obtained by the EnKF algorithm and more consistent with observations in central South American and central Africa, whereas the deviation in the EnKF method can be from -4 $m^2/m^2$ to 4 $m^2/m^2$. Furthermore, the assimilation shows better performance in the vegetation growing season. The lowest root-mean-square deviation (RMSD) is associated with the EAKF algorithm, suggesting that the EAKF algorithm is the best and has a robust performance.

The proportion of observations accepted by the land data assimilation system is another topic of this research. The proportion of accepted LAI observations is 10-20% in the low latitudes lower than in the high latitudes because of large biases between the assimilation and the observations. While low observation acceptance does not mean bad performance, indicating that assimilation result relies not only observation factor, but also the background error and ensemble model performance. When all the observations are accepted, the RMSD of the results is smaller than that when some observations are rejected.

The ensemble assimilation is conducted pointwise without considering spatial covariances, which will be considered in the future. Furthermore, more evolved techniques are needed to counteract the degeneracy of the particle filter.

*Code/Data availability*. The Community Land Model version 4.0 with carbon and nitrogen Components (CLM4CN) is a part of the Community Earth System Model version 1.1.1 (CESM1.1.1) developed by the National Center for Atmospheric Research (NCAR). The CESM code can be downloaded from http://www.cesm.ucar.edu/index.html. Developed and maintained by the Data Assimilation Research Section (DAReS) at NCAR, Data Assimilation Research Testbed (DART version lanai) can be downloaded from https://www.image.ucar.edu/DAReS/DART/.

*Author contributions*. All of the authors participated in the development of the paper's findings and recommendations.

*Competing interests*. The authors declare that they have no conflict of interest.

*Acknowledgments*. This work was jointly supported in part by the National Natural Science Foundation of China (2016YFA0600300 and 2017YFA0604300) and the Jiangsu Collaborative Innovation Center for Climate Change. Kevin Raeder (raeder@ucar.edu) is thanked for providing the DART/CAM4 reanalysis as ensemble meteorological forcing. Tim Hoar, Long Zhao and Yongfei Zhang are thanked for part of the coding and coupling with DART/CLM4CN.

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

**Table 1. Experimental design for LAI assimilation using DART/CLM4CN.**

| Experiment | Assimilated variables | Updated variables | Assimilation algorithm | Accept all observation |
|---|---|---|---|---|
| Algorithms | GLASS LAI | LAI, Leaf C, Leaf N | EAKF, EnKF, KF, PF | NO |
| Algorithms without observation rejection | GLASS LAI | LAI, Leaf C, Leaf N | EAKF | YES |

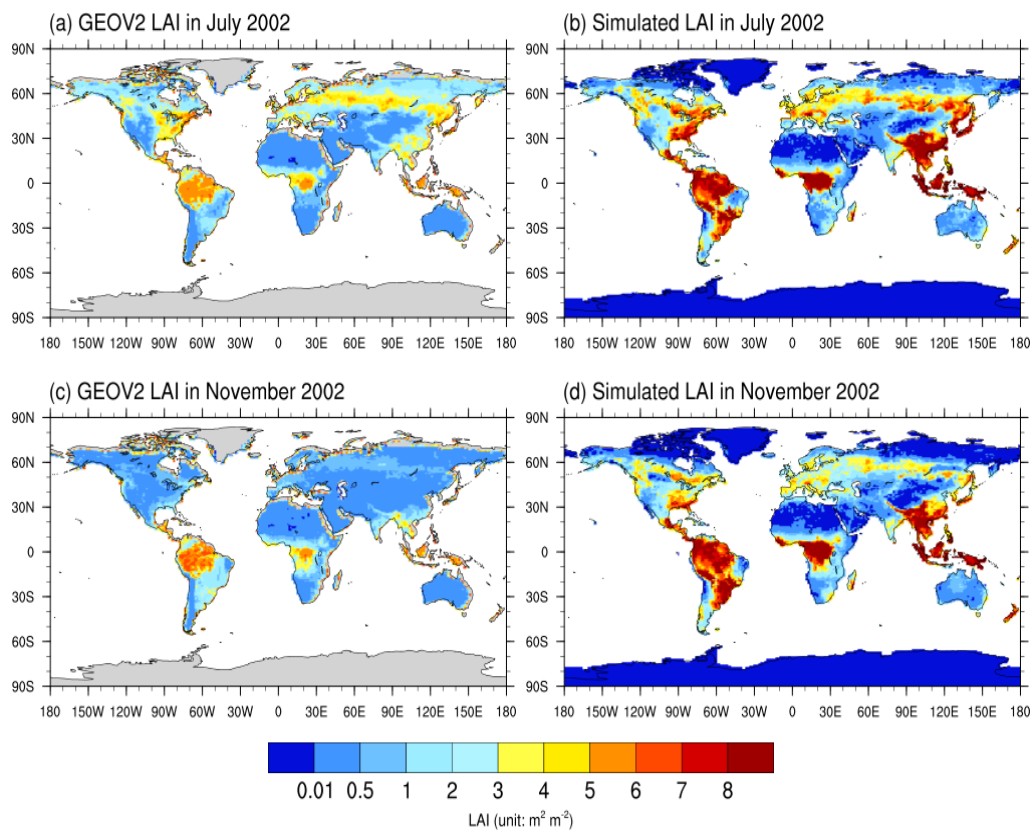

**Figure 1: Spatial distributions of global LAI values in 2002 for (a) GEOV2 LAI in July, (b) ensemble mean of simulations in July, (c) GEOV2 LAI in November, and (d) ensemble mean of simulations in November.**

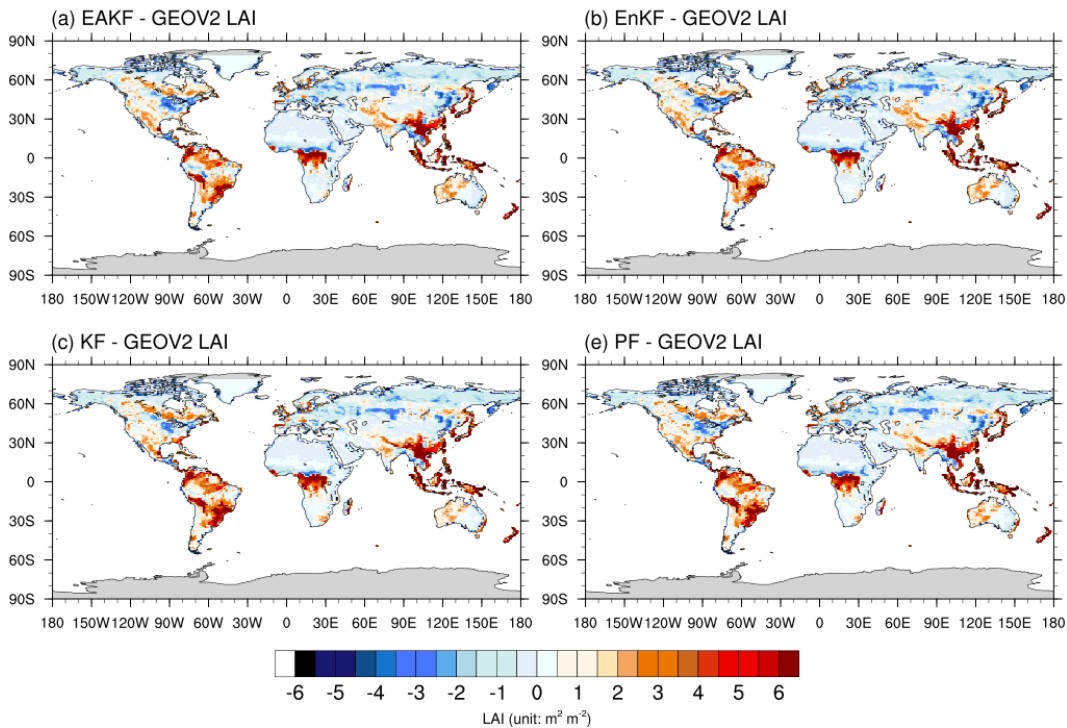

**Figure 2: Differences between global LAI from assimilation experiments with the methods of (a) EAKF, (b) EnKF, (c) KF and (d) PF and GEOV2 LAI in July 2002.**

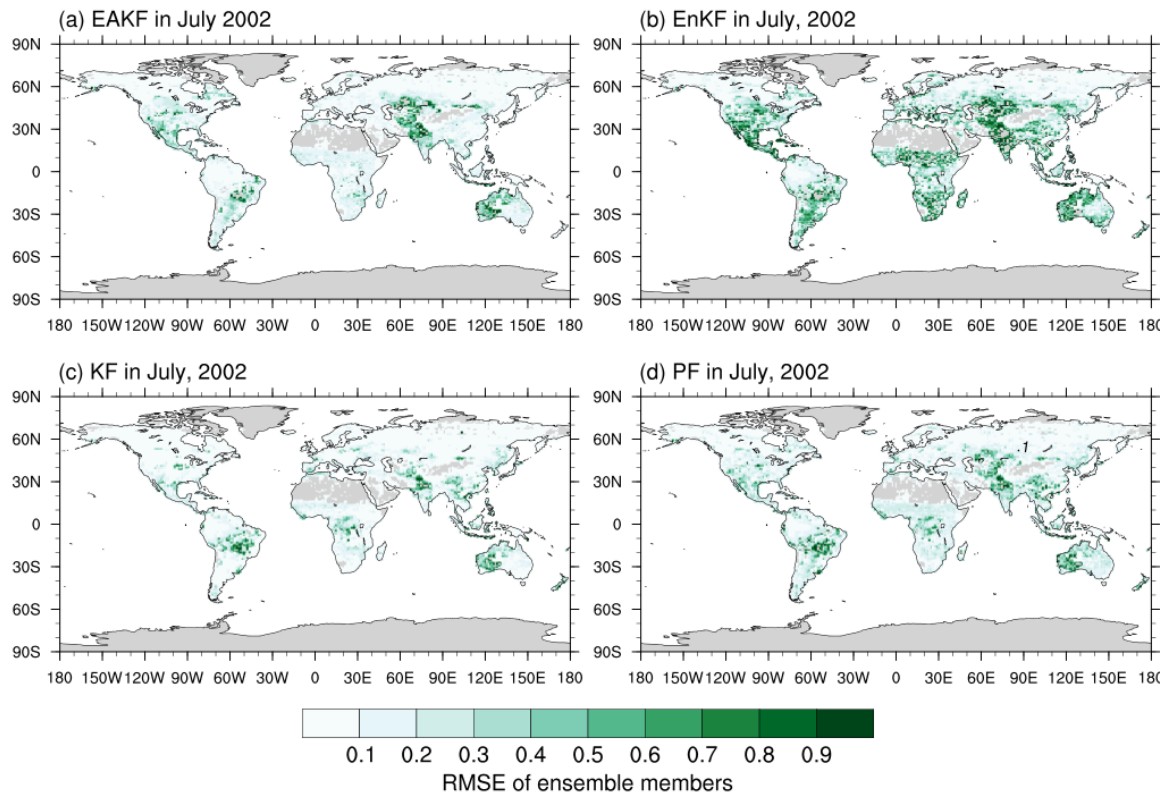

**Figure 3: Same as Fig. 2, but for RMSE of ensemble members.**

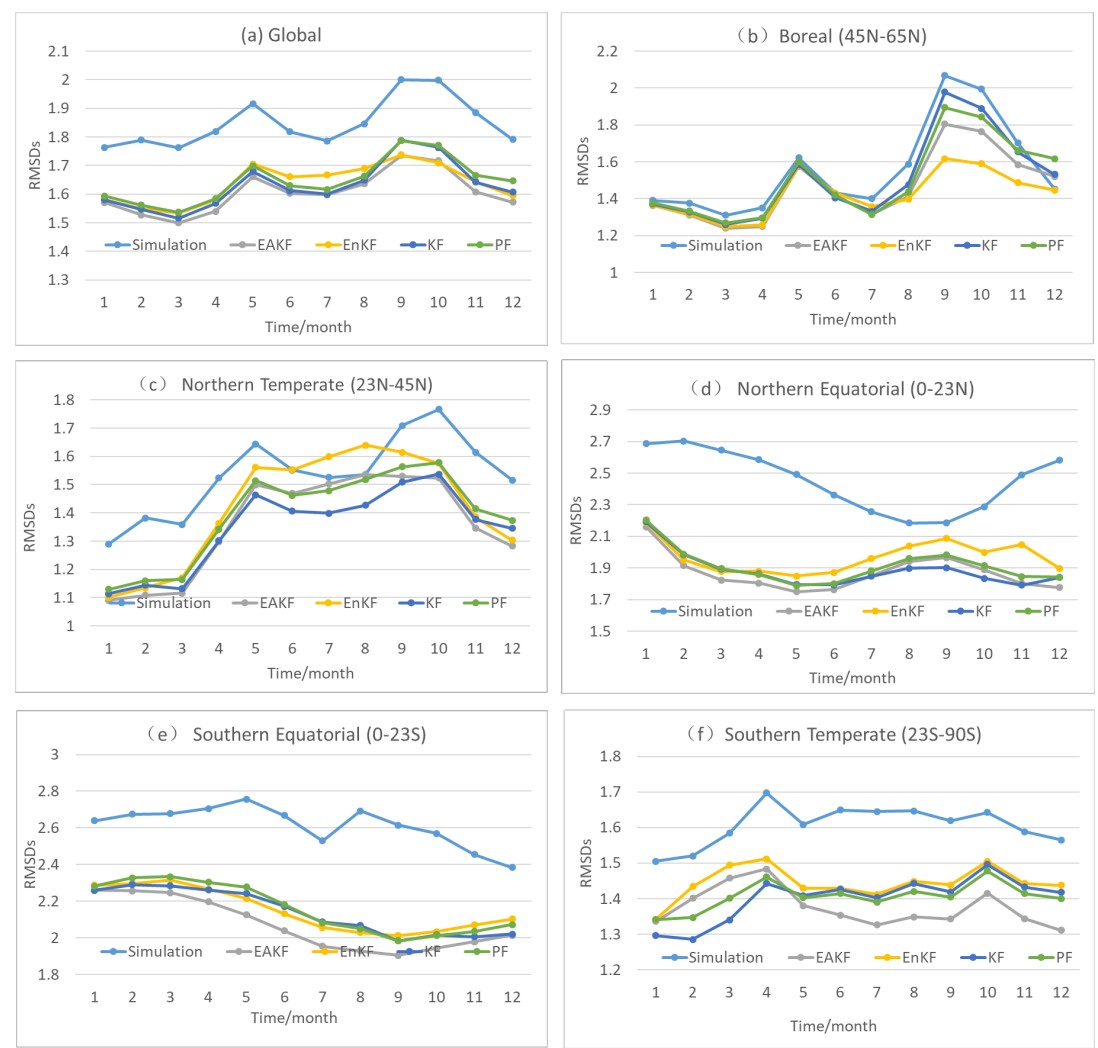

**Figure 4: RMSDs of ensemble means of simulation/assimilation versus GEOV2 LAI for (a) global, (b) boreal (45-65°N), (c) northern temperate (23-45°N), (d) northern equatorial (0-23°N), (e) southern equatorial (0-23°S), and (f) southern temperate (23-90°S).**

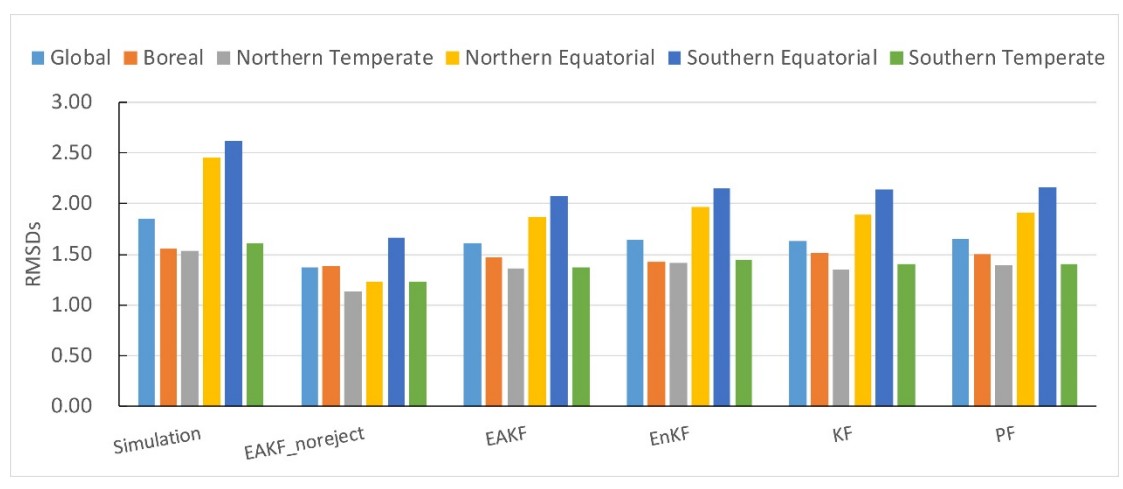

**Figure 5: Globally or regionally averaged RMSDs for the simulation/assimilation results and GEOV2 LAI.**

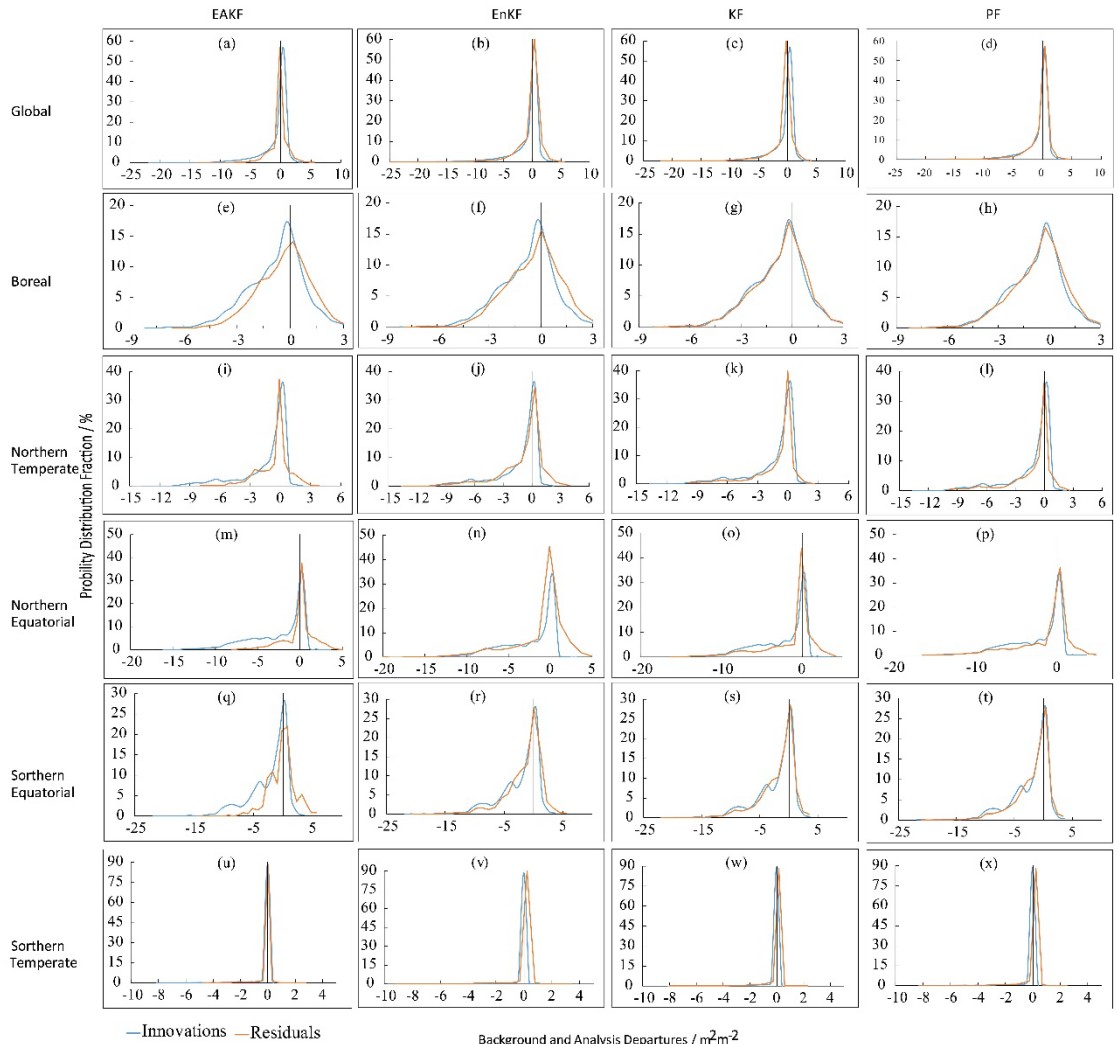

**Figure 6: The histograms of innovation and residuals of LAI globally and for all subregions during July 2002. (a-d) Global; (e-h) boreal; (i-l) northern temperate; (m-p) northern equatorial; (q-t) southern equatorial; (u-x) southern temperate**

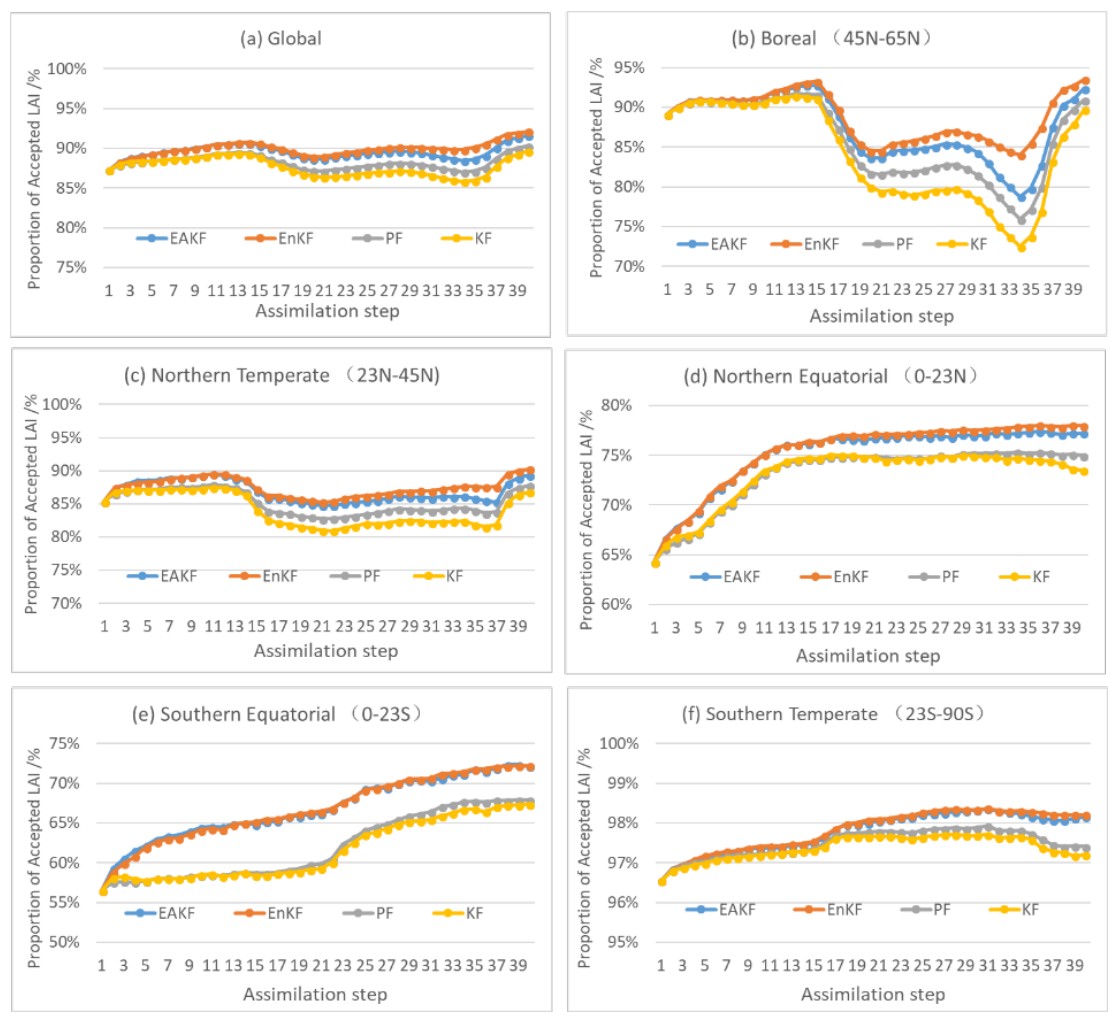

**Figure 7: The proportion of accepted LAI observations for the four algorithms in the zonal regions.**

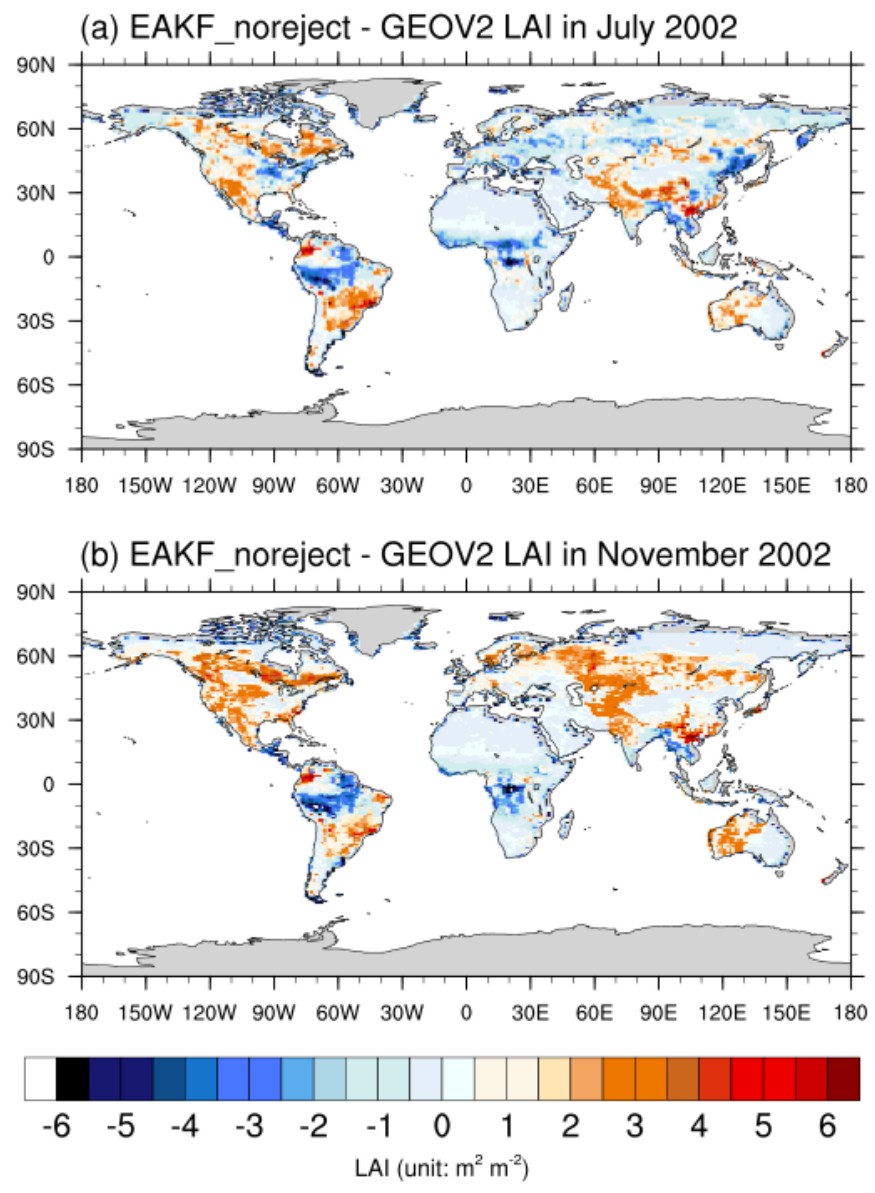

**Figure 8: Differences between globally assimilated and GEOV2 LAIs for the methods of EAKF in (a) July and (b) November.**

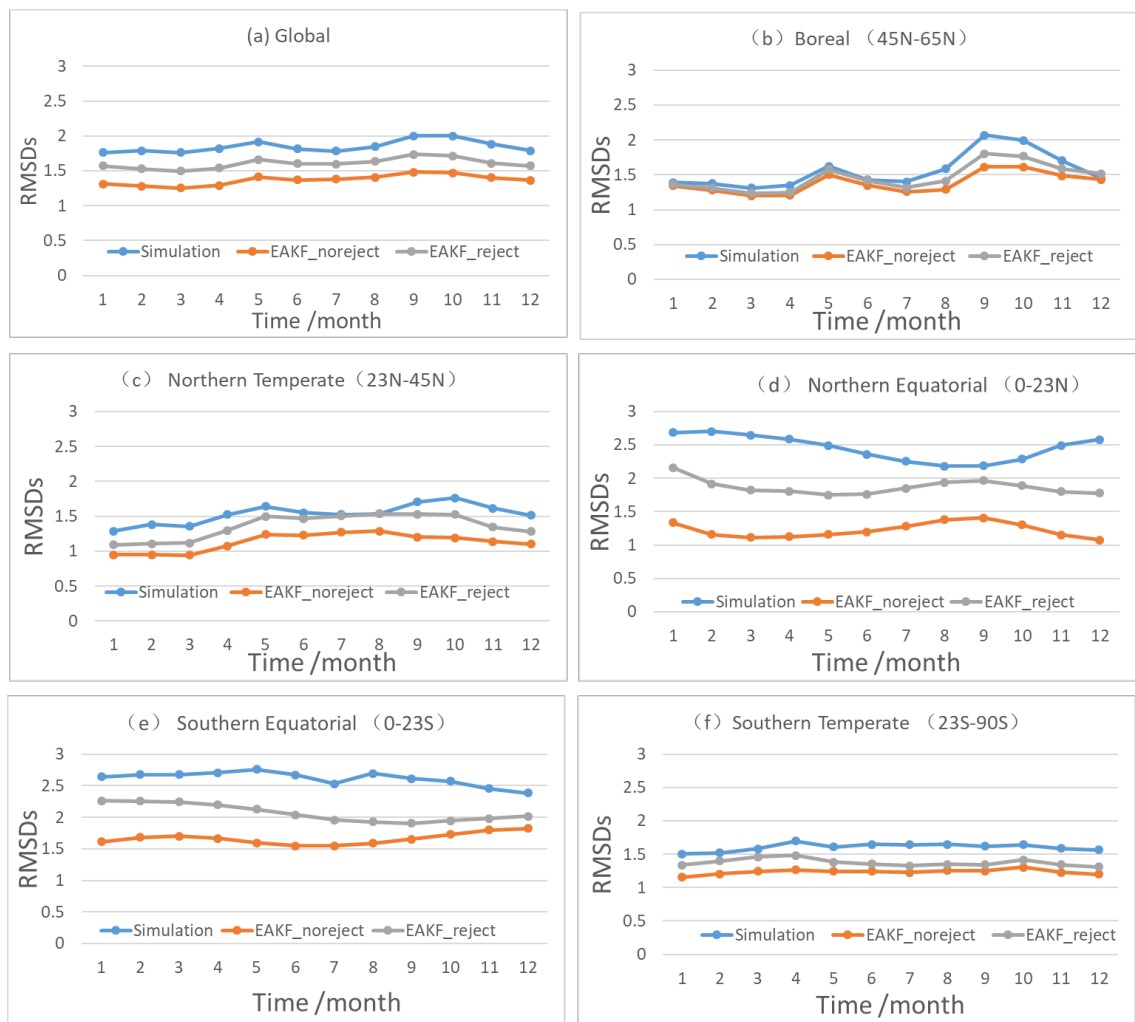

**Figure 9: RMSDs of simulation experiments without/with rejection (EAKF_noreject and EAKF_reject) and GEOV2 LAI for the (a) globe, (b) boreal (45-65°N), (c) northern temperate (23-45°N), (d) northern equatorial (0-23°N), (e) southern equatorial (0-23°S), and (f) southern temperate (23-90°S) regions.**