# Peer review of "Comparison of Different Sequential Assimilation Algorithms for Satellite-derived Leaf Area Index Using the Data Assimilation Research Testbed (Version Lanai)"

_Geoscientific Model Development, 2018_

## Referee Comment (RC1) · Anonymous Referee #1 · 25 Feb 2019

The submitted paper uses four assimilation methods (KF, EnKF, EAKF and PF) and CLM4CN to assimilate LAI, and chooses a best assimilation method by comparing with MODIS LAI. MODIS satellite remote sensing data can obtain LAI products with long time series. However, due to the impacts of cloud cover, aerosols, snow cover, and sensor failure, MODIS LAI products are characterized by high noise, low accuracy, and large fluctuations in the time series. Therefore, MODIS LAI data with better quality should be selected as observations based on quality control (QC) information. The research objective is reasonable and the review portion and figures need to be

improved.

1. what does the letter represent in formula (2)? It is not clear. 2. Line 13-15 in page 6, What method is used to solve the particle degradation problem in PF? 3. In section 2.4, time period of the atmospheric datasets is 1998-2010 in DA, why the time of LAI in the result is 2002? 4. What does "Observation Proportion" mean in Table? 5. Which version of MODIS LAI collection did you use? 6. There is no legend in Figure 1. Please add. 7. Due to the impacts of cloud cover, aerosols, snow cover, and sensor failure, MODIS LAI products are characterized by high noise, low accuracy, and large fluctuations in the time series. By calculating the RMSE of assimilation/simulation LAI and MODIS LAI, can this paper really choose a better assimilation algorithm? 8. Lines 2-3 in page 11, "assimilated observation" is mean "assimilated LAI"? 9. The legend and coordinate axis numbers are blurred in Figure 6. 10. "the distribution characteristics of both innovations and residuals are identical for the algorithms of KF and PF, which means that these two algorithms are not very efficient for LAI assimilation." Why innovations and residuals are identical, KF and PF are invalid. However, both innovations and residuals are not exactly the same for the algorithms of KF and PF ((g) and (h), (o) and (p) in Figure 6). 11. How to calculate the proportion of accepted LAI observations? 12. lines 3-4 in page 13, what are the conditions that observations are rejected during data assimilation. 13. lines 13-14 in page13, is RMSE calculated by EAKF_noreject/EAKF_reject and MODIS LAI?

---

## Referee Comment (RC2) · Anonymous Referee #2 · 25 Mar 2019

**1 OVERVIEW**

The paper proposes to compare the performance of four data assimilation (DA) algorithms in assimilating GLASS LAI within the CLM4CN land surface model (LSM) using the DART toolbox (version lanai). The four algorithms are: the Kalman filter (KF), an Ensemble Kalman Filter (EnKF), the Ensemble Adjustment Kalman Filter (EAKF) and a particle filter (PF). The authors show that the EAKF produces LAI estimates that are the closest to the assimilated observations. They also study the influence of observation

selection on LAI estimates compared to assimilated observations.

**2 GENERAL COMMENTS**

The objective of comparing assimilation methods for assimilating LAI in Land Data Assimilation Systems (LDASs) is fair and the choice of the various methods looks sound. The work belongs to a now long list of papers comparing DA methods in LDASs, most of them focusing on soil moisture. The novelty of the paper lies in the comparison of several DA methods assimilating LAI on global scale. Unfortunately the paper in its current form suffers from several issues that prevent it to be published as is. In particular:

- I think your results lack of analysis and validation. You only focus on assimilating GLASS LAI and compare newly LAI estimates with assimilated observations by computing RMSE. By using this sole criterion, you may miss something. The following analyses are missing:

  - The paper misses an analysis on the evolution of variances or ensemble spread of your LAI estimates.
  - You only focus on estimated LAI but your state vector also include Leaf C and Leaf N. How do these two variables evolve in time with DA?
  - You do not validate your approach with independent datasets. To validate a DA system, it is usual to compare control variables or other variables to independent datasets in order to check if assimilation has a positive impact. I suggest you use in-situ observations of LAI or use satellite estimates of evapotranspiration or gross primary production (estimates of both quantities have been shown improved by assimilating LAI) that are independent from the GLASS LAI product to validate your approach more thoroughly.

- Too many details in the description of the experimental setup are missing. For example:

  - Which period of time does your experiment cover? You have atmospheric forcing covering the period 1998-2010 but you only show results for the year 2002. Does that mean your experiment only cover one year? If so, this is not enough to determiner seasonal tendencies. Adding another year of experiment would reinforce your conclusions. If your experiment covers more than a year, please show results for the other years.

  - At which resolution do your run CLM4CN? In Figure 1, you show pictures at 1.0° resolution. Does that mean you run your LSM at the same resolution? Also, I thought that the GLASS LAI dataset was available at 0.05° resolution. Do you do interpolation in order to create the LAI you assimilate?

  - What kind of criterion do you use for observation selection? Is it when "the observed LAI is three times larger than the bias between the simulation and the observations" (l 16-17, p. 13)?

  I know it is impossible to include every detail in a paper or in supplementary materials. But I would like to remind the authors that every reader should be able to reproduce the experiment you conducted after reading a paper. In current form, your paper does not satisfy this important criterion.

- Too many details are also missing in the description of the DA methods you use.

  - I suspect your DA system works pointwise meaning you do not consider spatial covariances in KF, EnKF and EAKF. This is a strong hypothesis (perfectly respectable one). Could you confirm or reject my claim? If true, you should emphasize that point in your paper. If not, the whole analysis of spatial covariances is missing.

  - Could you recall in the paper the different equations involved for each DA method you use? Since it is a paper that compares various DA methods, the reader would benefit from having those written.

  - From what I read, it is impossible to determine which version of the particle filter you are using. Do you use the traditional Sequential Importance Resampling (SIR) filter from Gordon et al. (1993) or do you use more evolved techniques to counteract the degeneracy of the particle filter?

  - To run each member of your ensemble, you use 40 different atmospheric forcings selected from the 80-members DART/CAM4 dataset. How do you select them? Are they representative of the spread (uncertainty) of the whole 80-members atmospheric forcing dataset? If you select them randomly, you may have under-sampling issues (increasing the risk of filter divergence either for EnKF, EAKF and PF). Could you elaborate more on that subject?

  - Ensemble Kalman Filters (either what you call EnKF and EAKF) underestimate systematically variances. What do you do to counteract this problem? Do you use inflation (additive, multiplicative)? If so, how? If not, why?

As you can see the list of my comments is quite long. I do detail few of them in the next section. Nevertheless, I still consider the paper worth to be published if all points are addressed and, therefore, ask for a major revision.

**3  SPECIFIC COMMENTS**

- About the (lanai) in the title, could you make it more explicit that lanai is a version of DART in the title? It is confusing for the reader if she/he does not know what DART is.

- p. 1, l. 13-14, "*To improve the ability to simulate land surface water and energy balances*", since you show nothing related land surface water or energy fluxes, I suggest you to remove that comment.

- p. 1, l. 23, "*The PF algorithm performs worse than the EAKF and EnKF . . .*". You only consider RMSE as a criterion using for the PF the sampled mean. While using the mean makes sense for Ensemble Kalman Filters, for PF you have more freedom, one could use the particle with the biggest weight (a posteriori maximum for the pdf) for example. Could you add nuance to this statement?

- The introduction tends to mix general DA references to LDAS references making unclear for reading. I suggest you split your review in different paragraphs, one dedicated to DA in general, one dedicated to LDASs and one to the assimilation of LAI. Also many references are missing. Among others:

    – for DA in general: Bannister (2016), Vetra-Carvalho et al. (2018),
    – for LDASs: Lahoz and De Lannoy (2014), Reichle et al. (2014), De Lannoy et al. (2016), Sawada et al. (2015), Sawada (2018)
    – for assimilation of LAI: Sabater et al. (2008), Ines et al. (2013), Jin et al. (2018), Fox et al. (2018)

    Those references should help you build a thorough introduction.

- In section 2.2, can you recall that you use the lanai version of DART?

- Section 2.3.1 about the Kalman Filter (KF). The KF can only be used if your model is linear. Is your LSM linear between two times of observations (roughly 8 days)? If so please indicate what makes CLM4CN linear (as most LSMs are not!). If not, what you are using is rather an Extended Kalman Filter (EKF), in that case, how do you propagate the error covariance matrix from one time of observation to another i.e. how do you calculate the Jacobian matrix of your model?

- Section 2.3.2 about the Ensemble Kalman Filter. What you call the Ensemble Kalman Filter (EnKF) is likely the stochastic Ensemble Kalman Filter introduced by Burgers et al. (1998) and Houtekamer and Mitchell (1998) meaning that observations are perturbed for each member of the ensemble. Could you confirm it? And if so, please refer to those two papers.

- p. 5, l. 33. Eq (1) is false. The denominator of the fraction should be $\sigma_o^p + \sigma_{jo}^p$

- p. 6, l. 8. The variables involved in Eq. (2) are not defined.

- Section 2.5. You put Table 1 in section 2.5 but there is no mention in the text of the observation proportion you perform. Could you add sentences on that subject in section 2.5?

- p. 6, l. 29. You refer to the GLASS LAI dataset but afterwards you instead call them MODIS LAI. While I know GLASS LAI is from MODIS from 2002, it is rather confusing. Could you harmonize your notation?

- p. 7, Fig 1. There is no scale for Figure 1

- p. 8, l. 5-6. "*Figure 4 presents the root mean square errors (RMSEs) . . .*" Strictly speaking, they are not RMSEs but RMSDs (root-mean square differences) since your observations are not perfect. Please replace RMSE by RMSD.

- p. 10, Fig. 4 It looks like the assimilation is far less efficient in the boreal area than in other places. Can you explain why?

- p. 10, Fig 5. The RMSE for EnKF is not consistent to what is shown in Fig 4 (EnKF and EAKF give close results). Can you explain why?

- p. 11, Fig 6. I cannot read the figure. Can you make it bigger?

- p. 13, Fig 8. Have you compared LAI estimates (when you use observation selection) with every obs of LAI or only with those selected? It is rather normal that RMSDs are larger when you do not assimilate every observation than when you do. It would be worth comparing LAI estimates (when you use/do not use observation selection) with the selected observations only and see if you obtain smaller RMSDs.

**4  REFERENCES**

Bannister, R. N. A review of operational methods of variational and ensemble-variational data assimilation, Q. J. R. Meteorol. Soc., 143, 607–633 (2016).

Burgers, G., van Leeuwen, P. J. and Evensen, G. Analysis scheme in the ensemble Kalman filter, Mon. Wea. Rev., 126, 1719–1724 (1998).

Houtekamer, P. L. and Mitchell, H. L. Data assimilation using an ensemble Kalman filter technique, Mon. Wea. Rev., 126, 796–811 (1998).

De Lannoy, G. J. M., de Rosnay, P. and Reichle, R. H. Soil moisture data assimilation. In Handbook of Hydrometeorological Ensemble Forecasting, edited by Q. Duan, F. Pappenberger, J. Thielen, A. Wood, H. Cloke and J. C. Schaake. (2016).

Fox, A. M., Hoar, T. J., Anderson, J. L., Arellano, A. F., Smith, W. K., Litvak, M. E., et al. Evaluation of a data assimilation system for land surface models using CLM4.5. Journal of Advances in Modeling Earth Systems, 10, 2471–2494 (2018).

Gordon, N. J., Salmond, D. J. and Smith, A. F. Novel approach to nonlinear/non-Gaussian Bayesian state estimation, IEE Proc., 140, 107–113 (1993).

Ines, A. V. M., Das, N. N., Hansen, J. P. and Njoku, E. G. Assimilation of remotely sensed soil moisture and vegetation with a crop simulation model for maize yield prediction, Remote Sensing of Environment, 138, 149–164 (2013).

Jin, X., Kumar, L., Li, Z., Xu, X., Yang, G. and Wang, J. A review of data assimilation of remote sensing and crop models, Eur. J. Agron., 92, 141–152 (2018).

Lahoz, W. A. and De Lannoy, G. J. M. Closing the Gaps in Our Knowledge of the Hydrological Cycle over Land: Conceptual Problems, Surv. Geophys., 35, 623–660 (2014).

Reichle, R. H., De Lannoy, G. J. M., Forman, B. A., Draper, C. S. and Liu, Q. Connecting Satellite Observations with Water Cycle Variables Through Land Data Assimilation: Examples Using the NASA GEOS-5 LDAS, Surv. Geophys., 35, 577–606 (2014).

Sabater, J. M., Rüdiger, C., Calvet, J.-C., Fritz, N., Jarlan, L. and Kerr Y.: Joint assimilation of surface soil moisture and LAI observations into a land surface model, Agr. Forest Meteorol., 148, 1362–1373 (2008).

Sawada, Y., Koike, T. and Walker, J. P. A land data assimilation system for simultaneous simulation of soil moisture and vegetation dynamics. J. Geophys. Res. Atmos., 120, 5910–5930 (2015).

Sawada, Y. Quantifying Drought Propagation from Soil Moisture to Vegetation Dynamics Using a Newly Developed Ecohydrological Land Reanalysis, Remote Sens., 10, 1197 (2018).

Vetra-Carvalho, S., van Leeuwen, P. J., Nerger, L., Barth, A., Altaf, M. U., Brasseur, P. Kirchgessner, P. and Beckers, J.-M. State-of-the-art stochastic data assimilation methods for high-dimensional non-Gaussian problems, Tellus A, 70, 1445364 (2018).

---

## Author Comment (AC1) · 11 May 2019

**Major comments**

The submitted paper uses four assimilation methods(KF, EnKF, EAKF and PF) and CLM4CN to assimilate LAI, and chooses a best assimilation method by comparing with MODIS LAI. MODIS satellite remote sensing data can obtain LAI products with long time series. However, due to the impacts of cloud cover, aerosols, snow cover, and sensor failure, MODIS LAI products are characterized by high noise, low accuracy, and large fluctuations in the time series. Therefore, MODIS LAI data with better quality should be selected as observations based on quality control (QC) information. The research objective is reasonable and the review portion and figures need to be improved.

**Response:**Thanks very much for your comments to improve this manuscript. In the revised manuscript, we have focused on the following issues. 1. A thorough proofreading for language has been done to this manuscript, also the quality of all the figures has been improved.

2. The description for the experimental design and spin-up process has been added in Section 2. The ensemble simulation during the time period from 1998 to 2001 was treated as spin-up process, which can interpret why the result was shown for the year of 2002.

3. The datasets for assimilation and estimation was also included in Section 2.4.2. The Global Land Surface Satellite (GLASS) LAI datasets was used as the assimilated observation. To evaluate the assimilation result, an improved LAI dataset developed from the MODerate Resolution Imaging Spectroradiometer (MODIS) was utilized, which can reduce the spatial and temporal inconsistencies observed at the local spatial or temporal scales by considering the characteristics of the MODIS LAI data and quality control (QC) information

**Specific comments**

1. What does the letter represent in formula (2)? It is not clear.

   **Response:** If there are enough observations, the posterior density at k can be approximated

   $$p(X_k^a|Y_{1:k}) \approx \sum_{n=1}^{N} w_{i,k}\delta(X_k^a - X_{i,k}^a)$$

   in which $\delta(*)$ is the Dirac Function and $\sum_{n=1}^{N} w_{i,k} = 1$. $p(X_k^a|Y_{1:k})$ is the posterior probability distribution, $X_{i,k}^a$ is the particle element, $w_{i,k}$ is the weight of each particle, N is the number of particles.

2. Line 13-15 in page 6, What method is used to solve the particle degradation problem in PF?

   **Response:** We didn't do anything to solve the particle degradation in this study, maybe in the future we could focus on this topic.

3. In section 2.4, time period of the atmospheric datasets is 1998-2010 in DA, why the time of LAI in the result is 2002?

   **Response:** The 80 atmospheric forcing datasets with 6-hour time intervals for the period of 1998-2010 were used in this study. Actually only 40 members was randomly selected by considering computational cost and filter performance. The reason for the time of LAI in the result is 2002 is listed as follows. Firstly, the ensemble simulation during the time period from 1998 to 2001 was treated as spin-up process. We may miss the section of description for the spin-up process, which has been added in Section 2.4.1. Secondly, the purpose of this study is to find out the optimal algorithm, meaning that many experiments will be designed. Aiming at global scale, only one-year assimilation and ensemble simulation were conducted in considering of computational cost. We were trying to firstly find out the best experiment, and then conducting a long-term simulation or assimilation in the future.

4. What does "Observation Proportion" mean in Table 1?
**Response:** Sorry for the confusion. We have changed the word from "Observation Proportion" to "Algorithms without observation rejection". We also add some details for this kind of experiments in Section 2.5.

5. Which version of MODIS LAI collection did you use?

    **Response:** Global Land Surface Satellite (GLASS) LAI datasets was used in this study as assimilated observation (Zhao et al., 2013). As the ensemble simulation or assimilation was run at a resolution of 0.9° latitude by 1.25° longitude, the original spatial resolution of 0.05° of GLASS LAI is upscaled to the same resolution. To evaluate the assimilation result, an improved LAI dataset developed from the MODerate Resolution Imaging Spectroradiometer (MODIS) (Yuan et al., 2011) was utilized, which can reduce the spatial and temporal inconsistencies observed at the local spatial or temporal scales by considering the characteristics of the MODIS LAI data and quality control (QC) information (Baret et al., 2013). The resolution is 1 kilometer, and was also upscaled to grid levels to evaluate the analysis LAI and assimilation effect. We also added section 2.4.2 during this revision.

6. There is no legend in Figure 1. Please add.

    **Response:** Figure 1 is improved in this revision.

7. Due to the impacts of cloud cover, aerosols, snow cover, and sensor failure, MODIS LAI products are characterized by high noise, low accuracy, and large fluctuations in the time series. By calculating the RMSE of assimilation/simulation LAI and MODIS LAI, can this paper really choose a better assimilation algorithm?

    **Response:** To evaluate the assimilation result, an improved LAI dataset developed from the MODerate Resolution Imaging Spectroradiometer (MODIS) (Yuan et al., 2011) was utilized, which can reduce the spatial and temporal inconsistencies observed at the local spatial or temporal scales by considering the characteristics of the MODIS LAI data and quality control (QC) information (Baret et al., 2013). The resolution is 1 kilometer, and was also upscaled to grid levels to evaluate the analysis LAI and assimilation effect. It is better to evaluate the LAI

estimation by using in-situ observations, but it is not possible to do the estimation at global scale.

8. Lines 2-3 in page 11, "assimilated observation" is mean "assimilated LAI"?

   **Response:** Yes, and it has been changed as suggested.

9. The legend and coordinate axis numbers are blurred in Figure 6.

   **Response:** Figure 6 is corrected in this revision.

10. "the distribution characteristics of both innovations and residuals are identical for the algorithms of KF and PF, which means that these two algorithms are not very efficient for LAI assimilation." Why innovations and residuals are identical, KF and PF are invalid. However, both innovations and residuals are not exactly the same for the algorithms of KF and PF ((g) and (h), (o) and (p) in Figure 6).

    **Response:** The word of identical is changed to similar, and furthermore, Figure 6 was improved during this revision.

11. How to calculate the proportion of accepted LAI observations?

    **Response:** During assimilation, DART can calculate the number of non-assimilated observation when the difference of prior mean and observation is larger than 3 times of the expected value. The proportion of accepted LAI observations is defined as the number of accepted observations divided by the number of total observations.

12. lines 3-4 in page 13, what are the conditions that observations are rejected during data assimilation.

    **Response:** The "Algorithms" experiments would reject some observation under certain conditions using the KF, EnKF, EAKF, and PF algorithms. The expected value of the difference between prior mean and observation is $\sqrt{\sigma_{prior}^2 + \sigma_{obs}^2}$, in which $\sigma_{prior}$ and $\sigma_{obs}$ are the standard deviation of prior PDF and observation PDF respectively. DART will reject the observation if the bias of prior mean and observation is larger than 3 times of the expected value.

13. lines 13-14 in page13, is RMSE calculated by EAKF_noreject (EAKF_reject and MODIS LAI?

    **Response:** Yes, and it has been changed as suggested.

[Figure]

**Supplement:**

[revised manuscript text omitted]

30    types and sources, diagnostic routines, as well as new utilities. The detailed settings for DART can be found at https://www.image.ucar.edu/DAReS/DART/.

Currently, the coupled DART/CLM4 model has produced many reanalysis data for snow and soil moisture. It has been found that snow DA can improve temperature predictions, especially over the Tibetan Plateau, implying great implications for future land DA and seasonal climate prediction studies

35    (Lin et al., 2016). Furthermore, the coupled DART/CLM framework would be employed to assimilate other variables, such as LAI, from various satellite sources and ground observations (i.e., truly multimission, multiplatform, multisensor, multisource, and multiscale). Ultimately, this would allow earth system models to be constrained by all types of observations to improve model performance for seasonal and decadal prediction skills.

**2.3 Sequential Assimilation Algorithms**

According to Anderson et al. (2001), Equation (1) was used to express how new sets of observations modify the prior joint state conditional probability distribution available from predictions based on previous observation sets.

$$\mathbf{p}(\mathbf{z}_{t,k}|\mathbf{Y}_{t,k}) = \mathbf{p}(\mathbf{y}_{t,k}^o|\ \mathbf{z}_{t,k})\ \mathbf{p}(\mathbf{z}_{t,k}\ |\ \mathbf{Y}_{t,k-1})/\ \mathbf{p}(\mathbf{y}_{t,k}^o|\ \mathbf{Y}_{t,k-1}) \qquad (1)$$

in which $\mathbf{Y}_{t,k}$ is defined as the superset of all observation subsets, $\mathbf{y}_{t,k}^o$ is the $k$th subset of observations at time t, $\mathbf{z}_{t,k}$ is the joint state-observation vector for a given $t$ and $k$. In ensemble applications, there is generally no need to compute the denominator of (1). Four algorithms for approximating the product in the numerator of (1) are presented below, and detail information can refer to Anderson et al. (2001).

**2.3.1 Ensemble Kernel Filter (EKF)**

The kernel filter mechanism, first proposed by Lindgren et al. (1993) and developed in Anderson and Anderson (1999), was incorporated in the DART and can be extended to the joint state space. For detail calculation process can refer to Anderson et al. (2001). The kernel filter is potentially general, because the values and expected values of the mean and covariance and higher-order moments of the resulting ensemble are functions of high-order moments of the prior distribution. However, when applied to large models, the algorithm will come to computational efficiency issues.

**2.3.2 Ensemble Kalman Filter (EnKF)**

The KF algorithm has not been widely used because of computing limitations and the linear model error assumption. The EnKF was proposed based on a Monte Carlo approximation, for which the background error covariance is approximated using an ensemble of forecasts (Evensen, 1994). The EnKF algorithm can be utilized for nonlinear systems and can also reduce the computing requirement of DA (Burgers et al., 1998; Evensen, 2003; 2007).

The EnKF procedure is divided into two stages: prediction and analysis. (1) In the prediction stage, the ensemble forecast field is generated from the ensemble initial condition, and the error covariance matrix of the ensemble forecast is calculated. (2) In the analysis stage, the simulation of each member of the ensemble is updated using the covariance matrix of observation vector error and state vector error. The traditional EnKF was used in this study (Houtekamer and Mitchell1998), which were an ensemble of Kalman Filters, each using a different sample estimate of the prior mean and observations.

**2.3.3 Ensemble Adjust Kalman Filter (EAKF)**

Although the forms of expression are different, the proposed EnSRF (Whitaker et al., 2002) and EAKF (Anderson, 2001) are the same algorithm.

The difference between the EAKF and the traditional EnKF lies in the adjustment of the gain matrix to avoid filtering the divergence problem by increasing the premise of the analysis error covariance (Anderson, 2003, 2007; Wang et al., 2007). In the EAKF algorithm, ensemble observation members are calculated by the observation operator, and the increment of each observation member is calculated as $\Delta Y_i$.

The increment $\Delta X_{ij}$ for each ensemble sample of each state variable in terms of $\Delta Y_i$ can then be calculated as follows:

$$\Delta X_{ij} = \frac{\sigma_{jo}^p}{\sigma_o^p + \sigma_{jo}^p} \Delta Y_i. \tag{2}$$

where $i$ indicates the ensemble member, $j$ is the state vector member, $\sigma_{jo}^p$ is the prior covariance of state vector and observation, and $\sigma_o^p$ is the prior variance of observation.

**2.3.4 Particle Filter (PF)**

The Particle Filter (PF) is also a sequential Monte Carlo method, which is based on the Bayesian sequential importance sampling method (SIS). The PF algorithm finds a set of random samples in the state space to approximate the probability density function and then replaces the integral operation with the sample mean to obtain the process of minimum variance distribution of the state (Moradkhani et al., 2005). The procedure of the PF algorithm can also be divided into two frameworks: forecast and analysis.

If there are enough observations, the posterior density at k can be approximated as

$$p(X_k^a | Y_{1:k}) \approx \sum_{i=1}^N w_{i,k} \, \delta(X_k^a - X_{i,k}^a). \tag{3}$$

$\delta(*)$ is the Dirac Function and $\displaystyle\sum_{i=1}^N w_{i,k} = 1$.

in which $p(X_k^a | Y_{1:k})$ is the posterior probability distribution, $X_{i,k}^a$ is the particle element, $w_{i,k}$ is the weight of each particle, $N$ is the number of particles. Unlike the EnKF algorithm, the PF method takes into account the weights of different particles and can be better applied to nonlinear systems. However, in association with the DA, there are a limited number of particles with large weights, and too many computing resources are distributed to particles with weights of approximately 0. This situation is called particle degradation (Doucet et al., 2000). Effective methods to solve this issue include resampling or selecting more reasonable importance functions.

**2.4 Datasets**

**2.4.1 Ensemble Meteorological Forcing and initial conditions**

The ensemble initial conditions and background error (Hu et al., 2014) are produced from ensemble analysis products generated by running DART and the Community Atmosphere Model (CAM4) (Raeder et al., 2012). DART/CAM4 produced 80 atmospheric forcing datasets with 6-hour time intervals for the period of 1998-2010. These ensemble meteorological data have been widely employed in DA for ocean, snow, soil moisture, and many other related studies (Danabasoglu et al., 2012). By considering computational cost and filter performance, 40 members among the ensemble forcing datasets are chosen to drive the CLM4CN.

To achieve a steady state solution for all state variables, the CLM4CN was run for 4000 years by Qian's forcing (Qian et al., 2006) at a resolution of 1.9° latitude by 2.5° longitude (Shi et al., 2013). Then the CLM4CN was forced by the ensemble mean of selected 40 members of DART/CAM datasets for 1000 years. At last step, the ensemble simulation during the time period from 1998 to 2001 was treated as spin-up process, and 40 ensemble initial conditions were achieved. Aiming at global scale, only oneyear assimilation and ensemble simulation were conducted in considering of computational cost. We were trying to firstly find out the best experiment, and then conducting a long-term simulation or assimilation in the future, so only one year of assimilation results were presented.

**2.4.2 LAI datasets**

Global Land Surface Satellite (GLASS) LAI datasets was used in this study as assimilated observation (Zhao et al., 2013). As the ensemble simulation or assimilation was run at a resolution of 0.9° latitude by 1.25° longitude, the original spatial resolution of 0.05° of GLASS LAI is upscaled to the same resolution.

To evaluate the assimilation result, an improved LAI dataset developed from the MODerate Resolution Imaging Spectroradiometer (MODIS) (Yuan et al., 2011) was utilized, which can reduce the spatial and temporal inconsistencies observed at the local spatial or temporal scales by considering the characteristics of the MODIS LAI data and quality control (QC) information (Baret et al., 2013). The resolution is 1 kilometer, and was also upscaled to grid levels to evaluate the analysis LAI and assimilation effect.

**2.5 Experimental Design**

**Table 1.** Experimental design for LAI assimilation using DART/CLM4CN.

[revised manuscript text omitted]

---

## Author Response (AR1)

**Major comments**

The submitted paper uses four assimilation methods (KF, EnKF, EAKF and PF) and CLM4CN to assimilate LAI, and chooses a best assimilation method by comparing with MODIS LAI. MODIS satellite remote sensing data can obtain LAI products with long time series. However, due to the impacts of cloud cover, aerosols, snow cover, and sensor failure, MODIS LAI products are characterized by high noise, low accuracy, and large fluctuations in the time series. Therefore, MODIS LAI data with better quality should be selected as observations based on quality control (QC) information. The research objective is reasonable and the review portion and figures need to be improved.

**Response:** We appreciate your comments, which are helpful for us to further improve this paper. In the revised manuscript, we have focused on the following issues.

1. Proofreading has been done to improve the readability and quality of this manuscript. The quality of all the figures has also been improved.

2. The description for the experimental design and spin-up process has been added to Section 2. The ensemble simulation during the time period of 1998 ~ 2001 is treated as spin-up, which explains why the result is shown for the year 2002.

3. The datasets for assimilation and estimation are introduced in Section 2.4.2. The Global Land Surface Satellite (GLASS) LAI dataset is used as observations for assimilation. To evaluate the assimilation result, an improved LAI dataset developed from the MODerate Resolution Imaging Spectroradiometer (MODIS) is utilized, which can reduce the spatial and temporal inconsistencies by considering the characteristics of the MODIS LAI data and quality control (QC) information

**Specific comments**

1. What does the letter represent in formula (2)? It is not clear.

**Response:** If there are enough observations, the posterior density at $k$ can be approximated by

$$p(X_k^a|Y_{1:k}) \approx \sum_{i=1}^{N} w_{i,k}\, \delta(X_k^a - X_{i,k}^a)$$

in which $\delta(*)$ is the Dirac Function and $\sum_{i=1}^{N} w_{i,k} = 1$. $p(X_k^a|Y_{1:k})$ is the posterior probability distribution, $X_{i,k}^a$ is the particle element, $w_{i,k}$ is the weight of each particle, $N$ is the number of particles.

2. Line 13-15 in page 6, What method is used to solve the particle degradation problem in PF?

**Response:** We didn't do anything to solve the particle degradation problem in this study. We will address this issue in our future studies.

3. In section 2.4, time period of the atmospheric datasets is 1998-2010 in DA, why the time of LAI in the result is 2002?

**Response:** 80 atmospheric forcing datasets at 6-hour intervals over the period of 1998-2010 are used in this study. Considering computational cost and filter performance, only 40 members are randomly selected. The reasons why the time of LAI in the result is 2002 are given below. First, the ensemble simulation during the time period of 1998 ~ 2001 was treated as spin-up. The description of the spin-up process has been added to Section 2.4.1. Second, the purpose of this study is to find out the optimal algorithm, which needs many experiments to be conducted. Aiming at global scale and considering the computational cost, only one-year assimilation and ensemble simulation are conducted. We try to first find out the best experiment, and then conduct long-term simulation or assimilation in the future.

4. What does "Observation Proportion" mean in Table 1?

**Response:** We apologize for the confusion. The phrase "Observation Proportion" has been changed to "Algorithms without observation rejection". We also add some details related to this type of experiments to Section 2.5.

5. Which version of MODIS LAI collection did you use?

**Response:** Global Land Surface Satellite (GLASS) LAI dataset is used in this study as observations for assimilation (Zhao et al., 2013). Since the ensemble simulation or assimilation is run at the resolution of 0.9° latitude by 1.25° longitude, the original

spatial resolution of 0.05° of GLASS LAI is upscaled to the same resolution. To evaluate the assimilation result, an improved LAI dataset developed from the MODerate Resolution Imaging Spectroradiometer (MODIS) (Yuan et al., 2011) is utilized, which can reduce the spatial and temporal inconsistencies by considering the characteristics of the MODIS LAI data and quality control (QC) information (Baret et al., 2013). The resolution of MODIS LAI is 1-km, which is upscaled to grid level to evaluate the analysis of LAI and assimilation effect. Section 2.4.2 is newly added to the revised manuscript.

6. There is no legend in Figure 1. Please add.

**Response:** Figure 1 is improved and legend is added to the revised manuscript.

7. Due to the impacts of cloud cover, aerosols, snow cover, and sensor failure, MODIS LAI products are characterized by high noise, low accuracy, and large fluctuations in the time series. By calculating the RMSE of assimilation/simulation LAI and MODIS LAI, can this paper really choose a better assimilation algorithm?

**Response:** To evaluate the assimilation result, an improved LAI dataset developed from the MODerate Resolution Imaging Spectroradiometer (MODIS) (Yuan et al., 2011) is utilized, which can reduce the spatial and temporal inconsistencies by considering the characteristics of the MODIS LAI data and quality control (QC) information (Baret et al., 2013). The resolution is 1-km, which is upscaled to the grid level to evaluate the analysis of LAI and assimilation effect. It is better evaluate the LAI estimation by using in-situ observations, but it is not possible to do so on global scale.

8. Lines 2-3 in page 11, "assimilated observation" is mean "assimilated LAI"?

**Response:** You are right. The sentence has been changed as suggested.

9. The legend and coordinate axis numbers are blurred in Figure 6.

**Response:** Figure 6 is corrected in the revised manuscript.

10. "the distribution characteristics of both innovations and residuals are identical for the algorithms of KF and PF, which means that these two algorithms are not very efficient for LAI assimilation." Why innovations and residuals are identical, KF and PF are invalid. However, both innovations and residuals are not exactly the same for the algorithms of KF and PF ((g) and (h), (o) and (p) in Figure 6).

**Response:** The word "identical" is changed to "similar"; furthermore, Figure 6 has been improved in the revised manuscript.

11. How to calculate the proportion of accepted LAI observations?

**Response:** During assimilation, the DART can calculate the number of non-assimilated

observations when the difference of the prior mean and observations is larger than three times of the expected value. The proportion of accepted LAI observations is defined as the number of accepted observations divided by the number of total observations.

12. lines 3-4 in page 13, what are the conditions that observations are rejected during data assimilation.

**Response:** The "Algorithms" experiments would reject some observations under certain conditions using the KF, EnKF, EAKF, and PF algorithms. The expected value of the difference between the prior mean and observations is $\sqrt{\sigma_{prior}^2 + \sigma_{obs}^2}$, in which $\sigma_{prior}$ and $\sigma_{obs}$ are standard deviations of prior PDF and observation PDF respectively. DART will reject the observation if the bias of the prior mean and observation is larger than three times of the expected value.

13. lines 13-14 in page13, is RMSE calculated by EAKF_noreject/EAKF_reject and MODIS LAI?

**Response:** correct. The sentence has been changed as suggested.
The paper proposes to compare the performance of four data assimilation (DA) algorithms in assimilating GLASS LAI within the CLM4CN land surface model (LSM) using the DART toolbox (version lanai). The four algorithms are: the Kalman filter (KF), an Ensemble Kalman Filter (EnKF), the Ensemble Adjustment Kalman Filter (EAKF) and a particle filter (PF). The authors show that the EAKF produces LAI estimates that are the closest to the assimilated observations. They also study the influence of observation selection on LAI estimates compared to assimilated observations.

**2 GENERAL COMMENTS**

The objective of comparing assimilation methods for assimilating LAI in Land Data Assimilation Systems (LDASs) is fair and the choice of the various methods looks sound. The work belongs to a now long list of papers comparing DA methods in LDASs, most of them focusing on soil moisture. The novelty of the paper lies in the comparison of several DA methods assimilating LAI on global scale. Unfortunately the paper in its current form suffers from several issues that prevent it to be published as is. In particular:

• I think your results lack of analysis and validation. You only focus on assimilating GLASS LAI and compare newly LAI estimates with assimilated observations by computing RMSE. By using this sole criterion, you may miss something. The following analyses are missing:

1. – The paper misses an analysis on the evolution of variances or ensemble spread of your LAI estimates.

**Response:** Thank you for your suggestion. The RMSEs of the ensemble members are showed in Figure 3 to provide the hints where the assimilation is the most efficient. Please see Figure 3.

2. – You only focus on estimated LAI but your state vector also include Leaf C and

Leaf N. How do these two variables evolve in time with DA?

**Response:** In the former experiment, considering the large file size and limited storage capacity, we only output LAI. In the future, we can re-run the ensemble assimilation or simulation and output more variables if the storage capacity is increased.

3.  – You do not validate your approach with independent datasets. To validate a DA system, it is usual to compare control variables or other variables to independent datasets in order to check if assimilation has a positive impact. I suggest you use in-situ observations of LAI or use satellite estimates of evapotranspiration or gross primary production (estimates of both quantities have been shown improved by assimilating LAI) that are independent from the GLASS LAI product to validate your approach more thoroughly.

**Response:** To evaluate the assimilation result, an improved LAI dataset developed from the MODerate Resolution Imaging Spectroradiometer (MODIS) (Yuan et al., 2011) is utilized, which can reduce the spatial and temporal inconsistencies by considering the characteristics of the MODIS LAI data and quality control (QC) information (Baret et al., 2013).

• Too many details in the description of the experimental setup are missing. For example:

4.  – Which period of time does your experiment cover? You have atmospheric forcing covering the period 1998-2010 but you only show results for the year 2002. Does that mean your experiment only cover one year? If so, this is not enough to determiner seasonal tendencies. Adding another year of experiment would reinforce your conclusions. If your experiment covers more than a year, please show results for the other years.

**Response:** 80 atmospheric forcing datasets at 6-hour intervals over the period of 1998-2010 are used in this study. Considering the computational cost and filter performance, only 40 members are randomly selected. The reasons why the time of LAI in the result is 2002 are given below. First, the ensemble simulation during the time period of 1998 ~ 2001 is treated as spin-up. A detailed description of the spin-up process has been added to Section 2.5 in the revised manuscript. Second, the purpose of this study is to find out the optimal algorithm, which means that many experiments need to be conducted. Aiming at global scale and considering the computational cost, only one-year assimilation and ensemble simulation are conducted. We try to first find out the best experiment, and then conduct long-term simulation or assimilation in the future.

5.  – At which resolution do your run CLM4CN? In Figure 1, you show pictures at 1.0_

resolution. Does that mean you run your LSM at the same resolution? Also, I thought that the GLASS LAI dataset was available at 0.05_ resolution. Do you do interpolation in order to create the LAI you assimilate?

**Response:** The ensemble simulation or assimilation is run at the resolution of 0.9° latitude by 1.25° longitude. Therefore, the original spatial resolution of 0.05° of GLASS LAI is upscaled to the same resolution.

6. – What kind of criterion do you use for observation selection? Is it when "the observed LAI is three times larger than the bias between the simulation and the observations" (l 16-17, p. 13)?

**Response:** The expected value of the difference between the prior mean and observations is $\sqrt{\sigma_{prior}^2 + \sigma_{obs}^2}$, in which $\sigma_{prior}$ and $\sigma_{obs}$ are standard deviations of prior PDF and observation PDF, respectively. DART will reject the observation when the bias of prior mean and observation is larger than three times of the expected value. I know it is impossible to include every detail in a paper or in supplementary materials. But I would like to remind the authors that every reader should be able to reproduce the experiment you conducted after reading a paper. In current form, your paper does not satisfy this important criterion.

• Too many details are also missing in the description of the DA methods you use.

7. – I suspect your DA system works pointwise meaning you do not consider spatial covariances in KF, EnKF and EAKF. This is a strong hypothesis (perfectly respectable one). Could you confirm or reject my claim? If true, you should emphasize that point in your paper. If not, the whole analysis of spatial covariances is missing.

**Response:** We have further discussed this issue in the revised manuscript. Please see Section 2.5.

8. – Could you recall in the paper the different equations involved for each DA method you use? Since it is a paper that compares various DA methods, the reader would benefit from having those written.

**Response:** Thank you for your suggestion. We have recalled the equations in the revised manuscript.

9. – From what I read, it is impossible to determine which version of the particle filter you are using. Do you use the traditional Sequential Importance Resampling (SIR)

filter from Gordon et al. (1993) or do you use more evolved techniques to counteract the degeneracy of the particle filter?

**Response:** Following recommendations in the DART tutorial, the traditional Sequential Importance Resampling (SIR) filter from Gordon et al. (1993) is used in this study. Note that we didn't do anything to counteract the degeneracy of the particle filter.

10. – To run each member of your ensemble, you use 40 different atmospheric forcings selected from the 80-members DART/CAM4 dataset. How do you select them? Are they representative of the spread (uncertainty) of the whole 80-members atmospheric forcing dataset? If you select them randomly, you may have under-sampling issues (increasing the risk of filter divergence either for EnKF, EAKF and PF). Could you elaborate more on that subject?

**Response:** 40 different atmospheric forcing datasets are selected randomly. Considering the computational cost and the EAKF performance (e.g., Reichle et al., 2002; Zhang et al., 2014), it is not necessary to conduct the assimilation with 80 atmospheric forcing datasets. The ensemble atmospheric forcing should be designed identical for the four experiments for the purpose to find out the optimal algorithm. Furthermore, investigating uncertainties caused by different meteorological forcing datasets is beyond the scope of this study.

11. – Ensemble Kalman Filters (either what you call EnKF and EAKF) underestimate systematically variances. What do you do to counteract this problem? Do you use inflation (additive, multiplicative)? If so, how? If not, why?

**Response:** We didn't do any inflation because the objective of the present study is to compare the performance of different algorithms provided by DART under the same condition. For this reason, we use the default settings in DART except for the algorithm. As you can see the list of my comments is quite long. I do detail few of them in the next section. Nevertheless, I still consider the paper worth to be published if all points are addressed and, therefore, ask for a major revision.

**3 SPECIFIC COMMENTS**

1. • About the (lanai) in the title, could you make it more explicit that lanai is a version of DART in the title? It is confusing for the reader if she/he does not know what DART is.

**Response:** Thank you for your suggestion. We have changed the description from " DART (lanai)" to "DART (version Lanai)".

2. • p. 1, l. 13-14, "To improve the ability to simulate land surface water and energy balances", since you show nothing related land surface water or energy fluxes, I suggest you to remove that comment.

**Response:** As suggested, this sentence has been deleted.

3. • p. 1, l. 23, "The PF algorithm performs worse than the EAKF and EnKF : : :". You only consider RMSE as a criterion using for the PF the sampled mean. While using the mean makes sense for Ensemble Kalman Filters, for PF you have more freedom, one could use the particle with the biggest weight (a posteriori maximum for the pdf) for example. Could you add nuance to this statement?

**Response:** As suggested, we have added this statement to the revised manuscript.

4. • The introduction tends to mix general DA references to LDAS references making unclear for reading. I suggest you split your review in different paragraphs, one dedicated to DA in general, one dedicated to LDASs and one to the assimilation of LAI. Also many references are missing. Among others:

– for DA in general: Bannister (2016), Vetra-Carvalho et al. (2018),

– for LDASs: Lahoz and De Lannoy (2014), Reichle et al. (2014), De Lannoy et al. (2016), Sawada et al. (2015), Sawada (2018)

– for assimilation of LAI: Sabater et al. (2008), Ines et al. (2013), Jin et al. (2018), Fox et al. (2018)

Those references should help you build a thorough introduction.

**Response:** Thank you for your suggestion. The introduction has been improved in the revised manuscript. We also added many new references to this section, including those you mentioned.

5. • In section 2.2, can you recall that you use the lanai version of DART?

**Response:** The subtitle has been changed from "DART" to "DART (the Lanai version)". We also added some details to Section 2.2.

6. • Section 2.3.1 about the Kalman Filter (KF). The KF can only be used if your model is linear. Is your LSM linear between two times of observations (roughly 8 days)? If so please indicate what makes CLM4CN linear (as most LSMs are not!). If not, what you are using is rather an Extended Kalman Filter (EKF), in that case, how do you propagate the error covariance matrix from one time of observation to another i.e. how do you calculate the Jacobian matrix of your model?

**Response:** Thank you very much for your suggestion. Generally speaking, the CLM4CN is nonlinear, so the Kalman Filter could not be used for the LSMs. We have

checked the DART tutorial, and found that the algorithm we used in this study is the Ensemble Kernel Filter (EKF). We apologize for this mistake, and the detailed information about the EKF has been added to Section 2.3.1.

7. • Section 2.3.2 about the Ensemble Kalman Filter. What you call the Ensemble Kalman Filter (EnKF) is likely the stochastic Ensemble Kalman Filter introduced by Burgers et al. (1998) and Houtekamer and Mitchell (1998) meaning that observations are perturbed for each member of the ensemble. Could you confirm it? And if so, please refer to those two papers.

**Response:** As suggested, we have added this information to Section 2.3.2. The references are also added to the revised manuscript.

8. • p. 5, l. 33. Eq (1) is false. The denominator of the fraction should be $\sigma_o^p + \sigma_{j_o}^p$.

**Response:** Thank you for your information. We have corrected the equation.

9. • p. 6, l. 8. The variables involved in Eq. (2) are not defined.

**Response:** If there are enough observations, the posterior density at $k$ can be approximated by

$$p(X_k^a|Y_{1:k}) \approx \sum_{i=1}^{N} w_{i,k}\, \delta(X_k^a - X_{i,k}^a)$$

in which $\delta(*)$ is the Dirac Function and $\sum_{i=1}^{N} w_{i,k} = 1$. $p(X_k^a|Y_{1:k})$ is the posterior probability distribution, $X_{i,k}^a$ is the particle element, $w_{i,k}$ is the weight of each particle, $N$ is the number of particles.

10. • Section 2.5. You put Table 1 in section 2.5 but there is no mention in the text of the observation proportion you perform. Could you add sentences on that subject in section 2.5?

**Response:** We apologize for the confusion. We have changed the phrase from "Observation Proportion" to "Algorithms without observation rejection". We have also added some details related to this type of experiment to Section 2.5.

11. • p. 6, l. 29. You refer to the GLASS LAI dataset but afterwards you instead call them MODIS LAI. While I know GLASS LAI is from MODIS from 2002, it is rather confusing. Could you harmonize your notation?

**Response:** Global Land Surface Satellite (GLASS) LAI dataset is used in this study as observations for assimilation (Zhao et al., 2013). As the ensemble simulation or assimilation is run at the resolution of 0.9° latitude by 1.25° longitude, the original spatial resolution of 0.05° of GLASS LAI is upscaled to the same resolution. To

evaluate the assimilation result, an improved LAI dataset developed from the MODerate Resolution Imaging Spectroradiometer (MODIS) (Yuan et al., 2011) is utilized, which can reduce the spatial and temporal inconsistencies by considering the characteristics of the MODIS LAI data and quality control (QC) information (Baret et al., 2013). The resolution is 1-km, which is also upscaled to the grid level to evaluate the analysis of LAI and assimilation effect. We also added section 2.4.2 to the revised manuscript.

12. • p. 7, Fig 1. There is no scale for Figure 1

**Response:** Figure 1 has been improved in the revised manuscript.

13. • p. 8, l. 5-6. "Figure 4 presents the root mean square errors (RMSEs) : : :" Strictly speaking, they are not RMSEs but RMSDs (root-mean square differences) since your observations are not perfect. Please replace RMSE by RMSD.

**Response:** Thank you for your suggestion. All the RMSEs in this manuscript have been changed into RMSDs.

14. • p. 10, Fig. 4 It looks like the assimilation is far less efficient in the boreal area than in other places. Can you explain why?

**Response:** The assimilation is far less efficient in the boreal region than in other areas, which is partly attributed to the consistently low initial RMSD during non-growing seasons and limited capability of the model to simulate processes associated with boreal forest types.

15. • p. 10, Fig 5. The RMSE for EnKF is not consistent to what is shown in Fig 4 (EnKF and EAKF give close results). Can you explain why?

**Response:** There are some misunderstandings in Fig.5, in which the RMSD for EAKF is the value for the EAKF_noreject experiment, while the RMSDs for the other three algorithms are the ones from the reject experiments. We apologize for the confusion, and we have improved Fig. 5 in the revised manuscript. Furthermore, we have added new values to compare the difference of RMSDs between EAKF_noreject and EAKF_reject experiments, which are discussed in Section 4.

16. • p. 11, Fig 6. I cannot read the figure. Can you make it bigger?

**Response:** Figure 6 is corrected in this revision.

17. • p. 13, Fig 8. Have you compared LAI estimates (when you use observation selection) with every obs of LAI or only with those selected? It is rather normal that RMSDs are larger when you do not assimilate every observation than when you do. It would be worth comparing LAI estimates (when you use/do not use observation

selection) with the selected observations only and see if you obtain smaller RMSDs.

**Response:** Thank you for your suggestion. Figure 9 shows the RMSDs of simulation experiments without/with rejection (EAKF_noreject and EAKF_reject) and MODIS LAI for the globe and subregions. We have added details to revised manuscript.

**Response:** Thank you for the kind information. We have added these references to the revised manuscript.

[revised manuscript text omitted]

---

## Author Response (AR2)

**Topical Editor Decision: Publish subject to minor revisions (review by editor)**

**Comments to the Author:**

Dear Authors,

I assessed your response to all reviewers' comments and asked an additional opinion to one of the reviewers to make a decision on your manuscript. In general, you provide satisfactory answers to most comments, and prepared an improved version of the manuscript. However, I am concerned about the validation step of your analysis. Using a different MODIS-based LAI product is not really an independent dataset for validation. Reviewer 2 recommends the use of the CGLS LAI product (see comments below) for an independent validation. If you decide not to perform an additional validation, then I recommend you to abstain to mention that your DA system has been validated.

Regards,

Carlos A. Sierra

**Response to the Editor:**

Dear Editor:

We would like to submit our revised manuscript, "Comparison of Different Sequential Assimilation Algorithms for Satellite-derived Leaf Area Index Using the Data Assimilation Research Testbed (version lanai)", to the Geoscientific Model Development.

Thanks very much for your kind consideration on our manuscript. We appreciate the constructive comments and suggestions of the reviewers. We do not intend to abstain the comments of the second reviewer, and have performed an additional validation during this revision.

Following your and the 2nd reviewer's comments, the validated dataset has been changed from the MODIS LAI into the GEOV2 LAI from the Copernicus Global Land Service. The description for the validated dataset can be found in P7L11-P7L15.

Furthermore, all the figures relative to the observation (Fig.1, Fig.2, Fig.4, Fig.5, Fig.8 and Fig.9) have been corrected in the manuscript, as well as the relative description for all the figures. Fortunately, the main conclusions are still valid as before, indicating that our DA system has been validated.

Furthermore, a thorough proofreading for language has been done to the revised manuscript.

Best regards,

Xiaolu Ling

**Anonymous Referee #2**

**Suggestions for revision or reasons for rejection** (will be published if the paper is accepted for final publication)

The authors have significantly improved the manuscript. They have addressed adequatly most of my questions/comments/suggestions. In particular, experiments performed for this study are now fully detailed and can now be reproduced by any reader.
I still have an issue with the way the authors validate their data assimilation (DA) system. To validate a DA system, you are supposed to use observations that are independent from the assimilated observations. Unfortunately this is not the case here. The authors have assimilated LAI from the GLASS datasets based on MODIS data and then used another MODIS LAI dataset (Yuan et al., 2011) for validation. Since both datasets are derived from MODIS, they are not independent and, therefore, the validation is not correct. That is why I cannot accept the manuscript as is.
I suggest the authors validate their approach using datasets that are significantly independant from GLASS LAI. One possibility is to use LAI datasets from the Copernicus Global Land Service (CGLS). Fully accessible, CGLS LAI is derived from sensors from SPOT-VGT and PROBA-V satellites. While using a neural network algorithm trained on MODIS database in 2004, CGLS LAI should be independent enough to validate the approach on 2002.

**Response:** Thanks very much for your constructive suggestion. During this revision, the validated LAI data is changed from the upscaled MODIS LAI into the GEOV2 LAI from the Copernicus Global Land Service. The description for the validated dataset can be found in P7L11-P7L15. Furthermore, all the figures relative to the observation (Fig.1, Fig.2, Fig.4, Fig.5, Fig.8 and Fig.9) have been corrected in the manuscript, as well as the relative description for all the figures. Fortunately, the main conclusions are still valid as before, indicating that our DA system has been validated.

[revised manuscript text omitted]